# Clinical and Correlated Responses among Steroid Hormones and Oxidant/Antioxidant Biomarkers in Pregnant, Non-Pregnant and Lactating CIDR-Pre-Synchronized Dromedaries (*Camelus dromedarius*)

**DOI:** 10.3390/vetsci8110247

**Published:** 2021-10-21

**Authors:** Ragab H. Mohamed, Arafat Khalphallah, Ken Nakada, Enas Elmeligy, Dalia Hassan, Eman A. Ebissy, Rehab A. Ghandour, Sabry A. Mousa, Ahmed S. A. Hassaneen

**Affiliations:** 1Department of Theriogenology, Obstetrics, and Artificial Insemination, Faculty of Veterinary Medicine, Aswan University, Aswan 81528, Egypt; Ragabhasan2016@gmail.com; 2Division of Internal Medicine, Department of Animal Medicine, Faculty of Veterinary Medicine, Assiut University, Assiut 71526, Egypt; 3Department of Theriogenology, School of Veterinary Medicine, Rakuno Gakuen University, 582 Bunkyodai-Midorimachi, Ebetsu 069-8501, Japan; kenn@rakuno.ac.jp; 4Veterinary Teaching Hospital, Faculty of Veterinary Medicine, Assiut University, Assiut 71526, Egypt; enaselmeligy@yahoo.com; 5Department of Animal & Poultry Hygiene and Environmental Sanitation, Faculty of Veterinary Medicine, Assiut University, Assiut 71526, Egypt; daliaomran@aun.edu.eg; 6Department of Animal Health, Desert Research Center, Maratia, Cairo 11311, Egypt; eman.ebissy@yahoo.com; 7Department of Physiology, Faculty of Veterinary Medicine, Cairo University, Giza 12211, Egypt; rehabghandour1@gmail.com; 8Division of Internal Medicine, Department of Medicine and Infectious Disease, Faculty of Veterinary Medicine, Cairo University, Giza 12211, Egypt; 9Department of Theriogenology, Obstetrics and Artificial Insemination, Faculty of Veterinary Medicine, South Valley University, Qena 83523, Egypt; ahmed.hassaneen@vet.svu.edu.eg

**Keywords:** antioxidants biomarkers, camel fertility, clinical findings, lipid profiles indices, steroids hormones, oxidative stress, renal functions

## Abstract

Overproduction of free radicals is controlled by antioxidant defense mechanisms. These defense mechanisms are achieved by antioxidant enzymes such as catalase (CAT). The current study aimed to assess the changes in steroid hormones, oxidant/antioxidants biomarkers, lipid profiles/liver functions indices, renal function biomarkers and minerals metabolism in non-pregnant, lactating or pregnant one-humped she-camels (*Camelus dromedarius*) pre-synchronized with controlled internal drug releasing. The study also focused on the correlational relationships between steroid hormones and the oxidant/antioxidant biomarkers, lipid profiles and liver functions indices, renal functions and mineral metabolism in these she-camels. The study was conducted on apparently healthy dromedary she-camels (*n* = 60) during breeding season. A sexually active camel-bull was introduced to she-camels pre-synchronized with CIDR. Fifty to sixty days after natural mating, she-camels were examined for pregnancy. She-camels were divided into three main groups according to both pregnancy and lactation as following: pregnant (PREG, *n* = 38) which was kept as control one, non-pregnant and lactating (LACT, *n* = 8), and non-pregnant and non-lactating she-camels (NPREG, *n*= 14). Steroid hormones, i.e., progesterone (P_4_), estradiol (E_2_) and cortisol, oxidant indictors, i.e., malondialdehyde (MDA), antioxidant biomarkers, i.e., superoxide dismutase (SOD), total antioxidant capacity (TAC), CAT and reduced glutathione (GSH), lipid profiles indices, renal functions and related minerals were assessed. The present study confirmed the efficacy of using CIDR for synchronization in she-camels. Significant elevations in serum steroids hormones in PREG compare with LACT and NPREG. The highest concentrations of MDA as lipid peroxidation and oxidative stress indictors and lowest levels of antioxidant biomarkers except for SOD, i.e., TAC, CAT and GSH, were reported in PREG compared with LACT and NPREG. PREG showed the highest liver enzymes activities and lowest total protein values. Remarkable increases in serum concentrations of renal function parameters and phosphorous (P) were observed in PREG when compared with the other two groups. The investigated she-camels revealed significant correlation between steroid hormones and the oxidant biomarkers, antioxidant biomarkers, liver functions, renal functions and minerals metabolism parameters. P_4_ showed positive correlations with antioxidant biomarkers, i.e., TAC, CAT and GSH, and serum aspartate aminotransferase (AST) activities, whereas negative correlations were reported between P_4_ and renal functions biomarkers, i.e., blood urea nitrogen (BUN), creatinine (Cr) and creatinine kinase (CK), and minerals metabolism parameters, i.e., P and magnesium (Mg), in CIDR pre-synchronized she-camels. In contrast, E_2_ and cortisol showed negative correlations with antioxidant biomarkers, i.e., TAC, CAT and GSH, lipid profiles/liver functions indices, i.e., AST, alkaline phosphatase (ALP) and γ-glutamyl transferase (GGT), CK and Mg, however, positive correlations were demonstrated between E_2_ and cortisol, and MDA, Cr and P in investigated she-camels. In conclusion, the present study confirmed the efficacy of using CIDR for synchronization in she-camels. The highest MDA levels as indictors for oxidative stress and the lowest antioxidant levels, i.e., TAC, CAT and GSH, except for SOD in pregnant she-camels, were attributable to physiological oxidative stress as excellent compensatory responses observed in the PREG group to face such a physiologic stage. Moreover, lower P levels in non-pregnant she-camels would be contributed to failure of conception or early embryonic death. The investigated she-camels revealed significant correlations between steroid hormones and the oxidant indicators, antioxidant biomarkers, lipid profile indices and renal functions biomarkers that provided better understanding for physiological stress during pregnancy in camels.

## 1. Introduction

Dromedary one-humped camels (*Camelus dromedarius*) are highly adapted to extreme environments in a wide range of arid, semi-arid and tropical areas in Asia, Africa and Australia [1,2]. These dromedary camels are important as multipurpose animals as they are used for meat, milk and races as well as being working animals [3].

Dromedary camels have poor reproductive performance; this is likely contributed to by seasonality, a prolonged inter-calving period [2], and they have been considered as short-day seasonal breeder animals where the reproductive cyclicity was disturbed during the summer season [4], because of high temperature as a main factor that negatively affects reproductive cyclicity during low breeding season [4]. The breeding season of dromedary camels extends from December to the end of February or May, in Egypt [5,6]. Dromedary she-camels have a unique estrous cycle which consists of a follicular wave pattern without a luteal phase if ovulation is not induced by mating [7].

Several trials have been performed to improve the reproductive performance of both female and male dromedary camels, and to control the follicular waves pattern-estrous cycle in she-camels as well [8,9,10,11,12]. Recently, controlled-internal drug release (CIDR) was successfully used for synchronization in dromedary she-camels [9,11,12].

Oxidative stress has been the condition in which there was insufficient antioxidants to combat the overproduction of oxidants such as free radicals or reactive oxygen species (ROS) [13]. Physiological stress was accompanied by increased oxygen requirements [14], which in turn enhanced the overproduction ROS. The oxidative destruction of lipid (lipid peroxidation) as one of the important consequences of oxidative stress releasing malondialdehyde (MDA) as the stable end-product in comparison to short lifespan free radicals [15]. Overproduction of free radicals is controlled by antioxidants defense mechanisms. These defense mechanisms were achieved enzymatically by antioxidant enzymes such as catalase (CAT) [16].

Oxidative stress is commonly known as a cellular or individual level imbalance between oxidants and antioxidants. Oxidative damage is considered as one result of such an imbalance and involves oxidizing cellular macromolecules, cell death by necrosis or apoptosis, as well as damage to the structural tissue. Oxidants are compounds which can oxidize target molecules, and this can be achieved through one of three actions: hydrogen atom abstraction, electron abstraction or oxygen addition [17].

MDA, which is the last lipid peroxidation product, enzymatic antioxidants such as superoxide dismutase (SOD), glutathione peroxidase and CAT, non-enzymatic antioxidants such as decreased glutathione, total antioxidant capacity (TAC) and total oxidant status (TOS) are widely used to estimate oxidant stress [18]. The occurrence of oxidative stress was reported under various conditions, such as mastitis, enteritis, sepsis, respiratory and joint disease, transport and pregnancy in ruminants, and oxidative stress should be reported for the prognosis as well as effective treatment of these conditions [19].

Generally, antioxidants delay and prevent oxidative damage [20]. Several studies focused on the role of TAC and SOD as antioxidant biomarkers [21,22,23] whereas the body was supported with different antioxidants to overcome the toxic effects of these ROS. Superoxide anions which were generated during metabolic processes were reduced to hydrogen peroxide in the presence of SOD [22,23].

More specifically, glutathione reduced (GSH) is the most well-known natural low-molecular-weight antioxidant. It can play a pro-oxidant role, which has a lesser extent than its antioxidant role [24], while CAT is one of the first line defense antioxidant enzymes. Long-term oxidative stress caused a continuous increase in the MDA and decrease in the level of antioxidants as well as antioxidant enzymes such as GSH and CAT after transient increase to combat and alleviate the toxic effects of oxidant substances [15,25,26].

Dramatic metabolic changes during pregnancy resulted in alteration of the biochemical parameters. Few studies reported the differences in the biochemical parameters between pregnant and non-pregnant dromedary she-camels [27,28]. The liver had a critical role in the metabolism of carbohydrate and protein [29]. Aspartate aminotransferase (AST), alkaline phosphatase (ALP), γ-glutamyl transferase (GGT) and albumins were the markers used for assessment of liver function [30].

Camelids had very powerful mechanisms in urea recycling. Camels could recycle up to 90% of BUN, in contrast to ruminants, which presented a value of 10 to 30%. Recycling of nitrogen in camels increased in the case of lower proteins in diet and/or dehydration [29]. Camel had very particular anatomical structures in the kidney [31]. Only a small volume of water was lost during the elimination of urea by the production of concentrated urine [32]. The blood levels of Cr, CK and BUN as waste products removed by the kidneys had been indicators for renal function [33,34]. Serum biochemical tests have been used to provide valuable information regarding the health condition and physiological status of animals [35,36].

To our knowledge, limited data were available regarding the relation between steroid hormones, biochemical parameters, oxidant and antioxidants parameters and both pregnancy and lactation in dromedary she-camels. The current study aimed to assess the efficacy of controlled-internal drug releasing and its relationship with conception rates and clinical changes one-humped she-camels. It also focused on the changes in steroid hormones, i.e., P_4_, E_2_ and cortisol, oxidant/antioxidants biomarkers, i.e., MDA as an indicator of lipid peroxidation and oxidative stress, and SOD, TAC, CAT and GSH as antioxidants, lipid profiles/liver functions indices, renal function biomarkers and mineral metabolism indicators in non-pregnant, lactating or pregnant one-humped camels (*Camelus dromedarius*) pre-synchronized with controlled internal drug releasing. The study also focused on the correlational relationships between steroid hormones and the oxidant/antioxidant biomarkers, lipid profiles and liver functions indices, renal functions and mineral metabolism in these she-camels. Based on the overmentioned background and the aim of the study, our experimental hypothesis is that different pregnancy and lactation statuses affect the level of steroid hormones, oxidant/antioxidant biomarkers, liver functions and kidney functions biomarkers, and mineral metabolism in CIDR pre-synchronized she-camels.

## 2. Materials and Methods

### 2.1. Ethical Guidelines

All animal procedures performed in this study were performed according to Institutional Animal Care and Use Committee guidelines. During the experiment, the animals were handled according to the Use and Animal Care Committee of Aswan University and Assiut University which basically agree with the *Guide for the use and care of Laboratory Animals of the National Institutes of Health*—USA (NIH publication No. 86–23, revised 1996).

### 2.2. Animals and Therapeutic Strategy 

The present study included 60 apparently healthy dromedary she-camels reared at Mariut Farms, El-Amria, Alexandria, Egypt during breeding season. Their mean body weight was 450 kg (range: 350–580 kg) and their mean ages were 10.6 years (range: 8–14 years). Camels were housed in an open yard. Animals were group-fed on diet composed mainly of commercial concentrates mixture (12% crude protein and 70% Total Digestible Nutrients; TDN) (4 kg\head\day) in addition to roughage material, which was Egyptian clover hay in summer and Egyptian clover in winter (10 kg\head\day). Drinking water was offered all day. All camels were examined to verify and exclude the disease or pathology of genital organs. All animals were reared in the same corrals; in an open yard of 1300 m^2^ area under the same circumstances. Each head had a space allowance of more than 20 m^2^. More details about the management are explained in Figure 1.

For the intra-vaginal insertion of controlled internal drug release (CIDR), the perineal regions of the she-camels were washed and disinfected. CIDR (containing 1.38 g P_4_, Pfizer^®^, Pfizer Animal Health, Division of Pfizer Inc., New York, NY 10017, USA) was gently inserted in the vagina of she-camel in standing position. The CIDR was left for 14 days. The observation showed none of the camels lost the CIDR. An experienced sexually active camel-bull of good fertility (with good breeding history) was introduced to she-camels after the removal of CIDR. The sexual desire of the used fertile camel-bulls was examined by showing rutting behaviour when introduced to she-camel (s) in heat. Fifty to sixty days after natural mating, the pre-synchronized mated she-camels were examined for pregnancy diagnosis using SonoAce R2 ultrasound scanner (Medison, Samsung, South Korea). She-camels were divided into three main groups according to both pregnancy and lactation; pregnant she-camels group (PREG, *n* = 38) which was kept as control one, non-pregnant and lactating she-camels (LACT, *n* = 8), and non-pregnant and non-lactating she-camels (NPREG, *n*= 14).

### 2.3. Study Location

Dromedary she-camels used in this study were reared at Mariut farms, El-Amria, Alexandria, Egypt. Alexandria governorate, Egypt, where the study was conducted, is located at −1 m above mean sea level, latitude 31.21° N and longitude 29.95° E. The study was performed during the period from December to May when the low (minimum) and high (maximum, °C) temperature ranged between 10–17 and 20–27 °C, respectively. The relative humidity ranged between 66% in May and 68% in December. The photoperiod throughout the whole period of the study extended from 10:15 h in December to 13:30 h in May.

Alexandria weather data (December to May) including maximum temperature (T, °C), and relative humidity (RH, %) were used to determine the temperature-humidity index (THI) (Figure 2) using the following equation [37]:THI=(1.8×T+32)−[(0.55−0.0055×RH)×(1.8×T−26)]

The threshold value of heat stress in she-camels was set as 72-points THI, where the stress was categorized into mild (72–79 THI), moderate (80–89 THI) and severe stress (≥90 THI) [37].

### 2.4. Samples

Blood samples were collected from the jugular vein at same time-point of ultrasonography (at the end of the breeding season, 2 months after the natural mating) into vacutainer tubes with sodium (Na) fluoride for determination of glucose levels in plasma and plain vacutainer tubes for separation of sera and measuring different biochemical parameters. Serum and plasma samples were collected and kept frozen at −20 °C for subsequent hormonal and biochemical analyses using commercial test kits according to the standard protocols of suppliers.

### 2.5. Clinical Examination

The clinical examinations included measurements of heart and respiratory rates and recording of rectal temperatures as well as rumen movements was performed as described by Fowler [38]; Abdel-Rahman et al. [39]; Hassan et al. [40].

### 2.6. Hormonal Analysis

Serum P_4_ concentrations were measured by enzyme-linked immunosorbent assay (ELISA-Sandwich Protocol) using Oxford Biomedical Research commercial kits, Rochester Hills, Michigan, USA. Serum cortisol and E_2_ concentrations were measured using commercial radioimmunoassay kits (Parameters commercial kits, R&D Systems, Inc. 614 McKinley Place NE Minneapolis, MN 55413, Toll Free USA, Canada).

### 2.7. Oxidant and Antioxidants Biomarkers Assays

The Spectro Ultraviolet-Vis RS spectrophotometer (Labomed, Inc., Los Angeles, CA, USA) was used to determine the serum concentrations of MDA colorimetrically as lipid peroxidation indicators (Biodiagnostic commercial kits, Cairo, Egypt), and serum concentrations of antioxidants biomarkers kinetically (Biodiagnostic assay kits, Cairo, Egypt) including SOD, TAC, CAT and GSH.

### 2.8. Liver Functions and Lipid Profile Indices

The Spectro Ultraviolet-Vis RS spectrophotometer (Labomed, Inc., Los Angeles, CA, USA) was used to determine blood concentrations of glucose, total protein, albumin, liver enzymes, i.e., AST, GGT and ALP. Serum globulin was determined by subtraction of albumin from total protein. Moreover, all kits and reagents were obtained from Gamma Trade Company (Cairo, Egypt) for plasma glucose, BioMed commercial kits (Cairo, Egypt) for total proteins and albumins, and Spectrum commercial kits (Cairo, Egypt) for AST, ALP and GGT.

### 2.9. Kidney Functions Biomarkers

The Spectro Ultraviolet-Vis RS spectrophotometer (Labomed, Inc., Los Angeles, CA, USA) was used to measure serum concentrations of BUN, Cr and CK using Bioscience commercial kits (Cairo, Egypt), BioMed commercial kits (Cairo, Egypt) and BioChain commercial kits (Bay Area, San Francisco, CA, USA), respectively.

### 2.10. Mineral Metabolism Indicators

The Spectro Ultraviolet-Vis RS spectrophotometer (Labomed, Inc., Los Angeles, CA, USA) was used to measure serum concentrations of calcium (Ca), phosphorus (P) and magnesium (Mg) were determined using BioMed commercial kits (Cairo, Egypt).

### 2.11. Statistical Analysis

Data were analyzed using SPSS statistical software program for Windows, version10.0.1 (SPSS Inc., Chicago, IL, USA). The obtained data were described as mean ± SD. The data obtained from the clinical findings and biochemical analyses were analyzed using one-way analysis of variance followed by Tukey’s multiple comparison whereas the significance level was set at *p* ≤ 0.05. The significant differences between the means of PREG compared with LACT and NPREG were evaluated. The response variables depicted a normal distribution. Correlation coefficient was calculated using Pearson Correlation at *p* < 0.05 or *p* < 0.01 between steroids hormones and the oxidant/antioxidant biomarkers, liver functions/lipid profiles indices, renal functions biomarkers and mineral metabolism indicators.

## 3. Results

### 3.1. Conception Rates and Clinical Parameters

In our study, 38 out of 60 healthy dromedary she-camels pre-synchronized with CIDR became pregnant with a conception rate of 63.33%.

The study reported no remarkable differences between the three groups of she-camels for temperature and rumen motility. They were within the reference ranges. In contrast, the other clinical findings including pulse and respiration showed significant elevations in PREG when their values were compared with those in LACT or in NPREG. The values for pulse and respirations were higher than the reported reference ranges. No significant changes were reported either for pulse or for respiration between LACT and NPREG (Table 1).

### 3.2. Serum Concentrations of Steroid Hormones

Pre-synchronized PREG she-camels showed a significant increase in serum concentrations of P_4_ compared with the other two groups. Serum concentrations of P_4_ were significantly lower in NPREG compared with LACT. Serum levels of P_4_ were higher than their reference ranges (Table 2).

Serum E_2_ concentrations were significantly higher in PREG when their values compared with those either in LACT or in NPREG. These significant differences were demonstrated between the two non-pregnant groups where serum E_2_ levels were remarkably elevated in NPREG compared with LACT. Serum concentrations of E_2_ were higher than their reference ranges (Table 2).

Pregnant pre-synchronized she-camels showed a significant raise in serum cortisol concentrations when their values were compared with those either in LACT or in NPREG. Serum levels of cortisol were not remarkably changed between LACT and NPREG. With reference to their reference values, higher levels of cortisol hormones were mentioned in PREG while they were within the reference ranges in the other two investigated groups, LACT and NPREG (Table 2).

### 3.3. Serum Concentrations of Oxidants and Antioxidants Biomarkers

Regarding oxidants biomarkers, serum MDA concentrations were significantly higher in PREG than the other two groups. Serum MDA values in PREG were higher than their reference ranges. Serum MDA levels in LACT were not remarkably changed when compared with those in NPREG, moreover, they were within the physiological reference ranges (Table 3).

For antioxidants biomarkers, with exception of SOD, serum concentrations of estimated antioxidants biomarkers, i.e., TAC, CAT and GSH, were significantly lower in PREG than LACT and NPREG (Table 3). Except for GSH, serum concentrations of all measured antioxidant biomarkers in PREG through the current study were within their reported reference ranges. Serum GSH levels were higher than their reference values (Table 3).

Serum concentrations of SOD were remarkably increased in PREG when their values compared with those in the other groups of she-camels. No significant changes in serum SOD activities were reported between LACT and NPREG and so they were within their reference values (Table 3).

PREG showed a significant reduction in serum concentrations of TAC, CAT and GSH when their values were compared with those either in LACT or in NPREG. Significant variations in serum concentrations were also reported for these antioxidant biomarkers between LACT and NPREG, whereas serum levels of TAC, CAT and GSH were remarkably higher in NPREG when their concentrations compared with those in LACT (Table 3).

### 3.4. Serum Concentrations of Liver Functions and Lipid Profile Indices

Plasma concentrations of glucose showed no remarkable changes between different she-camel groups where they were within their reference ranges (Table 4).

A significant drop in serum values of total proteins and globulins were described in PREG and LACT when their values compared with their values in NPREG. These significant changes were not observed for albumins between different she-camel groups. Serum values of total proteins, albumins and globulins were within the physiological ranges (Table 4).

PREG revealed a remarkable elevation in serum activities of AST, ALP and GGT when their values were compared with those in LACT or in NPREG. Significant variations in serum activities of AST, ALP and GGT were reported between LACT and NPREG whereas their serum activities were remarkably reduced in NPREG when their values compared with those in LACT. Serum activities of liver enzymes as indicators of liver functions and lipid profiles were within the reference ranges (Table 4).

### 3.5. Serum Concentrations of Kidney Functions Biomarkers

She-camels in the PREG group showed significant increases in serum values of BUN, Cr and CK when their values were compared with those in LACT or in NPREG. These significant changes in renal function biomarkers were absent between LACT or in NPREG when their values compared with each other. Serum concentrations of BUN, Cr and CK in all examined she-camels were within the physiological reference values (Table 5).

### 3.6. Serum Concentrations of Mineral Metabolism

She-camels pre-synchronized using CIDR showed no remarkable differences in serum concentrations of the estimated minerals elements with except for P, whereas serum P values were remarkably higher in PREG than those in LACT or in NPREG. Serum P concentrations revealed no significant changes between LACT and NPREG. Serum concentrations of Ca, Mg and P were within their reference ranges either in PREG, in LACT or in NPREG (Table 6).

### 3.7. Correlation between Steroid Hormones and Oxidant/Antioxidant Biomarkers

Significant correlations were reported between steroid hormones and oxidant/antioxidant biomarkers (Table 7).

Positive correlations were demonstrated between P_4_ and all antioxidant indicators except for SOD, including TAC, CA and GSH. Negative correlation was reported between P_4_ and SOD. Except for SOD, negative correlations were demonstrated between E_2_ and cortisol, and all antioxidant biomarkers including TAC, CA and GSH. Positive correlation was reported between both E_2_ and cortisol, and SOD (Table 7).

Serum P_4_ and cortisol values were negatively correlated with MDA; however, positive correlation was described between serum E_2_ and serum MDA (Table 7).

Serum P_4_ was negatively correlated with E_2_ and cortisol, hence positive correlation was demonstrated between E_2_ and cortisol (Table 7).

### 3.8. Correlation between Steroid Hormones and Liver Functions/Lipid Profile Indices

Positive correlation was demonstrated between P_4_ and AST activities. However, no remarkable correlational relationships were reported in she-camels between P_4_ and all other liver functions/lipid profiles except for AST. Serum E_2_ values were significantly correlated with lipid profiles and liver functions indices whereas serum E_2_ values were negatively correlated with TP, globulins, AST, ALP and GGT. She-camels pre-synchronized using CIDR showed significant correlations between cortisol and liver functions biomarkers, whereas serum cortisol concentrations were negatively correlated with serum activities of AST, ALP and GGT (Table 8).

### 3.9. Correlation between Steroid Hormones and Renal Functions Biomarkers

Steroid hormones were significantly correlated with renal functions biomarkers in she-camels pre-synchronized using CIDR. Serum P_4_ levels were negatively correlated with serum BUN and Cr values. Moreover, both serum E_2_ and cortisol concentrations were correlated positively with Cr and negatively with CK (Table 9).

### 3.10. Correlation between Steroid Hormones and Minerals Parameters

Serum P concentrations were positively correlated with both serum concentrations of E_2_ and cortisol, hence, serum Mg values were negatively correlated with serum P_4_ levels, (Table 10).

## 4. Discussion

### 4.1. Clinical Examination and Conception Rates

The present study successfully elucidated the difference in biochemical, hormonal, oxidant and antioxidant status due to pregnancy or lactation in she-camels. Moreover, this study reported the relation between steroid hormones, oxidants and antioxidants in she-camels. On the other hand, the physiological conditions had more of a biochemical and hormonal effect rather than hematological indices in camels raised under traditional conditions [41]. In the current study, out of 60 CIDR pre-synchronized dromedary she-camels, 38 became pregnant with a conception rate of 63.33%. The high conception rate reported in the present study strongly support the good efficacy of using CIDR for synchronization in she-camel as recently reported [9,10]. The current study reported no remarkable differences between the three groups of she-camels for temperature and rumen motility. They were within the reference ranges reported by Hassan et al. [40] and Hamad et al. [42]. In contrast, the other clinical findings including pulse and respiration showed significant elevations in PREG as a control group when compared with LACT or in NPREG. Their values for pulse and respirations were higher than the reported reference ranges reported by Fowler et al. [38] and Hassan et al. [40].

### 4.2. Serum Concentrations of Steroid Hormones

The significantly higher P_4_ and E_2_ concentrations in the pregnant she-camels presented in this study agreed with Ayoub et al. [43] who reported that the concentrations of sex steroid hormones varied according to different physiological conditions with higher concentrations in the pregnant she-camels compared to the non-pregnant ones. Moreover, the source of high P_4_ reported in non-pregnant lactating she-camels in the present study would be either ovarian or adrenal. Moreover, luteal activity was found even in non-mated she-camels in some cases due to spontaneous ovulation, possibly enhanced by [43]. In contrast, other literature mentioned that serum estrogen and cortisol concentrations were elevated clearly before parturition and have coincided with a decrease in serum P_4_ values that might support their roles in triggering parturition in she-camels. Furthermore, it could be concluded that together with other parameters, cortisol, E_2_ and P_4_ could be used as good indicators to predict the time of parturition in she-camels [39]. Here, the current results showed that significant differences for P_4_ and E_2_ were also reported between LACT and NPREG, whereas serum concentrations of P_4_ were significantly lower and serum E_2_ levels were remarkably higher in NPREG compared with LACT. Abdel-Rahman et al. [39] and Agarwal et al. [44] added that P_4_ levels elevated remarkably in pregnant she-camels after conception and during the first and the second trimester of pregnancy, then dropped remarkably during the third trimester. Moreover, P_4_ levels in she-camels declined significantly during the periparturient period. Other articles revealed that estrogen was continuously secreted during pregnancy in she-camels [44], and their concentrations raised at mid-gestation, indicating continued follicular development during pregnancy [45]. In the present study, the highest levels of P_4_ have been reported during pregnancy in she-camels. These results were supported by the previous reports which added that, in she camels, serum concentrations of P_4_ were constantly low [46], and their concentration started to rise after mating. Following mating, at least one corpus luteum was formed that secreted a significant amount of P_4_. Camels showed serum P_4_ levels above 2.0 and 3.0 ng/mL at 20 and 30 days of mating should be considered as pregnant [47].

Cortisol plasma levels, as a stress indicator hormone, increased after transportation [48], hypoxemia in new-world camelids [49]. The present study revealed that pregnancy affected the cortisol concentrations in she-camels. It was previously noted that serum cortisol concentrations remain low till the mid-stage of gestation and then sharply decrease after two weeks of calving in spite of lactation [39]. The current work added that CIDR pre-synchronized pregnant she-camels showed a significant raise in serum cortisol hormones concentrations compared with LACT or NPREG while no remarkable changes were reported between LACT and NPREG. With reference to the reference values reported by Saeb et al. [50], higher levels of cortisol hormones were mentioned in PREG while they were within the reference ranges in the other two investigated groups of she-camels. Ebissy et al. [51] reported that serum concentrations of cortisol were markedly increased on the day of parturition which might be due to stress of parturition due to the elevation in the concentration of adrenocorticotropic hormone secretion from the fetal pituitary [52,53]. On the other side, Abdel-Rahman et al. [39] mentioned that blood cortisol levels were significantly improved in the third trimester and this elevation continued markedly during the prepartum period, reaching their maximum levels on the day of parturition. Thereafter they significantly declined during the few days after parturition to day 15 postpartum.

### 4.3. Serum Concentrations of Oxidants and Antioxidants Biomarkers

Abd El-Hamid et al. [23] reported that blood antioxidant indicators, i.e., TAC and MDA, did not change markedly among different pregnancy periods of one-humped she-camels. In contrast, activities of SOD were remarkably higher in the first more than the second trimester, hence the lowest value was reported in the third trimester of the pregnancy period. Regarding oxidants biomarkers, the current study mentioned more highly significant increases in serum MDA levels in PREG than in the other two she-camel groups. Serum MDA values in PREG were higher than their reference ranges reported by Saleh et al. [54]; El-Deeb and Elmoslemany [55]; El-Bahr and El-Deeb [56]. This was likely due to physiological oxidative stress during pregnancy, which leads to overproduction of free radicals due to lipid peroxidation [39]. Serum MDA levels were not remarkably changed between LACT and NPREG; moreover, they were within the physiological reference ranges. Abd El-Rahman et al. [39] described the changes in MDA during different stages of pregnancy and around calving where they reported significant elevations in serum MDA in the last trimester compared to the other stages of pregnancy and after parturition. Furthermore, serum MDA values were higher than the reference ranges throughout the pregnancy period in she-camels. Walsh [57] stated that lipid peroxidation occurred naturally in cells and thus free radicals’ production was considered a normal physiological process, however, increased free radicals’ production during pregnancy might be attributable to the stressful condition which led to many metabolic changes such as an increase in the basal metabolic rate. This increase in free radicals produced an increase in the oxidative stress due to increased lipid peroxidation [58]. Furthermore, Jarikre et al. [59] referred to the fact that the significant increase in the MDA level might be associated with the risk effect of cellular damage and inflammation, which was associated with bronchopneumonia and bronchointerstitial pneumonia, in addition to the destruction of epithelial cells and a fibrinous reaction resulting from vascular damage. Singh et al. [60] referred to the significantly higher values of MDA as an indicator of higher levels of oxidative stress in pregnant camels. Blood levels of CAT and GSH were more remarkably reduced in pregnant and lactating camels than in the control camels. GSH was a key role in scavenging t-butyl hydroperoxide, an agent which induced lipid peroxidation [61] and catalase with its catalytic activity, as the most important antioxidant enzymes that inactivate large amounts of oxidants by transformation of superoxide anion into hydrogen peroxide and water [62]. Among the biomarkers, MDA levels in control and lactating animals were similar but CAT and GSH were markedly reduced in lactating camels, it showed that antioxidant defenses (CAT and GSH) exhausted earlier than the rise in levels of lipid peroxidation (MDA) [60]. Abdel-Rahman et al. [39] mentioned that TAC and GSH levels were lower in the third trimester than others, whereas the levels were relatively lower prepartum than 15 days postpartum. Moreover, MDA concentrations were markedly elevated during the late pregnancy period as a result of the physiological stress of pregnancy. The present work revealed that with exception of SOD, serum concentrations of estimated antioxidants biomarkers, i.e., TAC, CAT and GSH, were significantly lower in PREG than LACT and NPREG. Except for GSH, serum concentrations of all measured antioxidant biomarkers in PREG through the current study were within their reference ranges reported by Shoieb et al. [22]; Saleh et al. [54]; El-Bahr and El-Deeb [56] and Kamr et al. [63]. Serum GSH levels were higher than their reference values mentioned by Shoieb et al. [22]. The previous reports revealed that the low values of total antioxidant could be linked to the adaptive capacity in order to optimize the consumption of oxygen and neutralization of free radicals [39]. Panda et al. [64] reported that plasma total antioxidants activities did not vary among different treatment groups at 60 days before calving and continued to decline from 30 days prepartum till calving. Regarding to the current study, significant variations in serum concentrations of these antioxidant biomarkers, i.e., TAC, CAT and GSH were also reported between LACT and NPREG whereas serum levels of TAC, CAT and GSH were remarkably higher in NPREG compared with LACT. On the other hand, serum GSH concentrations were remarkably low in late pregnancy and increased gradually till early lactation. That might be due to the activity of blood GSH that was elevated with increased lipid peroxidation, while GSH activity had a negative correlation with MDA production [15,39]. Therefore, an imbalance between increased production of ROS and reduced availability of antioxidant defense near parturition might aggravate oxidative stress and might contribute to periparturient disorders in dairy cows [14]. Moreover, Kamr et al. [63] stated significantly higher concentrations of MDA in diseased camels when compared to those in the healthy ones, and a significant reduction in the levels of TAC and CAT compared to the healthy group. Regarding the present study, decreased antioxidants (TAC and catalase) might be attributed to the cell protection consumption of those enzymes by preventing the initiation of peroxidization and production of final products, such as thiobarbituric acid reacting substances, that could cause serious cell damage [65].

According to the present work, significantly higher levels of serum SOD were observed in PREG pre-synchronized she-camels. These results were in agreement with Abd El-Hamid et al. [23] that reported significantly higher values of SOD in the second and first trimester in one-humped she-camels, respectively. In contrast, the previous studies mentioned that the activities of SOD, CAT and GSH were markedly decreased in infected camels compared with the control, whereas lipid peroxidation was remarkably elevated as reflected in higher serum MDA values in the serum of infected camels compared with the control [54,56]. The decrease in SOD and CAT activities might be owed to the consumption of antioxidants as free radical scavengers during oxidative stress in diseased camels [54]. Infection in camels resulted in inhibition of antioxidants, i.e., GSH and SOD [54,56]. The decrease in glutathione level in the serum of infected camels was reflected in the observed reduction in the activities of SOD and CAT [56]. The current work added that no significant changes in serum SOD activities were reported between LACT and NPREG and their values were within the reference values stated by Shoieb et al. [22]; El-Deeb and Elmoslemany [55].

### 4.4. Serum Concentrations of Liver Functions and Lipid Profile Indices

Plasma concentrations of glucose in the current work showed no remarkable changes between different groups, where they were within their reference ranges reported by Saeb et al. [50]; Islam et al. [66] and Abdalmula et al. [67]. Similar findings were mentioned by Omidi et al. [28] who reported no remarkable variations in blood glucose levels in pregnant she-camels compared to non-pregnant ones. Regarding to the present study, the unchanged blood glucose levels in pregnant and non-pregnant she-camels might be as a result of sufficient homeostasis in maintaining the concentration of blood glucose in a constant range. Moreover, there was no seasonal variation in sampling time [28]. In contrast, Ebissy et al. [51]; Tharwat et al. [53] demonstrated significant elevation in blood concentrations of glucose in she-camels at the time of parturition compared to their values at the peripartum period and this might be due to the stress of parturition and coincide with elevation of cortisol at this period. On the other hand, Saeed et al. [68] and Kelanmer et al. [69] found a significant drop in glucose concentration in periparturient dromedary camels compared to the post-calving period. These findings were owed to the effect of the gestational state on glucose values.

Estimation of some serum parameters such as total proteins, albumins and globulins in dromedary camels were important for assessing the physiological status and health of heavy pregnancy, parturition and postpartum periods [28]. Here, a significant drop in serum values of total proteins and globulins was described in PREG and LACT when their values were compared with their values in NPREG. These significant changes were not observed for albumins between different she-camel groups. In contrast, the previous studies stated that serum concentrations of the total proteins and albumins did not differ markedly either pre- or postpartum. However, globulin concentrations were remarkably increased at day 28 postpartum. This could be resulted from the formation of immunoglobulins [51,53]. Moreover, Abd El-Hamid et al. [23] stated that plasma total proteins and albumins were not affected by the pregnancy period. The current work mentioned that the serum values of total proteins, albumins and globulins in she-camels were within the physiological ranges reported by Islam et al. [66] and Mohamed and Hussein [70]. In contrast to our results, El-Tohamy et al. [71] reported that protein concentration increases with the advancement of pregnancy. Moreover, Abdel-Rahman et al. [39] said that total protein and albumin were more significantly decreased in the second and the third trimesters than the first one without marked alterations in globulin levels. On the other side, remarkably lower concentrations of serum albumins were reported in pregnant she-camels [39]. Saleh et al. [72] also reported a negative correlation between serum total proteins levels and pregnancy in Holstein cows. The decrease in serum total proteins as parturition approached might be attributed to the fact that the fetus synthesized all of its proteins from the amino acids that derived from the dam, and the fetus growth increased exponentially, reaching a maximum level, especially in muscles, during late pregnancy [73]. In contrast, Abdel-Rahman et al. [39] demonstrated that serum globulin showed no significant alterations around parturition. This result disagreed with most findings, such as as Janku et al. [74], because this period could be associated mainly with the production of colostrum and other immunological changes typical for the periparturient period.

Ebissy et al. [51] reported a marked elevation in serum AST, GGT and ALP at +14 postpartum in dromedary she-camels compared to their values in the prepartum period. This significant higher pattern continued for serum activity of GGT and ALP at +28 after parturition. These findings contradict with the current study in which PREG revealed a remarkable increase in serum activities of AST, ALP and GGT when their values were compared with those in LACT or in NPREG. Significant variations in serum activities of AST, ALP and GGT were reported between LACT and NPREG whereas their serum activities were remarkably reduced in NPREG. On the other hand, the previous reports which mentioned a significant drop in serum activity of GGT in late pregnant one humped she-camels [28] which might be attributable to involvement of glutathione to maintain antioxidative status of the entire body [75]. Serum activities of liver enzymes as indicators of liver functions and lipid profiles were within the reference ranges reported by Islam et al. [66]; Mohamed and Hussein [70] and Kaneko et al. [76] either in pregnant, lactating or non-lactating she-camels. Abd El-Hamid et al. [23] reported significantly higher activity of plasma ALP in the first and second trimesters compared with the third trimester of pregnancy in she-camels, while the value of AST activity did not differ markedly among different pregnancy trimesters.

### 4.5. Serum Concentrations of Kidney Functions Biomarkers

Cr and BUN are waste products removed from the blood by the kidneys. Creatinine is a breakdown product of creatine phosphate in the muscle. Serum Cr is used as a biomarker for assessment of renal functions [33,34]. High BUN values usually indicate disturbance of kidney functions; however, other factors might be involved in blood BUN levels [28]. The pregnant she-camels in the current study demonstrated significant increases in serum values of BUN, Cr and CK when compared to LACT or in NPREG. These significant changes in renal functions biomarkers were absent between LACT and NPREG. These results agreed with Omidi et al. [28], who reported significantly higher levels of BUN and Cr in pregnant one-humped camels compared to non-pregnant ones. This significant elevation in serum concentrations of BUN and Cr levels could be referred as a part of the homeorhetic mechanisms for camels’ adaptation during late pregnancy. The high urea-recycling rates in camels might transfer urea to the gastrointestinal tract as a source of “non-protein nitrogen” instead of being excreted as urine. Kidney health biomarkers must be cautiously tested especially in heavy pregnancy periods. Furthermore, Patel et al. [34] added that the quantity of Cr formed each day depended on the total body contents of creatine, which in turn depended on the dietary intake, rate of synthesis of creatine and muscle mass. In contrast to the current study, Ebissy et al. [51]; Tharwat et al. [53] reported no remarkable changes in concentration of BUN in the tested time points either pre- or postpartum. However, Cr concentrations were significantly decreased at the first week and the second week postpartum. The present study stated that serum concentrations of BUN, Cr and CK in all examined she-camels were within the physiological reference values reported by Saeb et al. [50]; El-Deeb and Elmoslemany [55] and Mohamed and Hussein [70].

### 4.6. Serum Concentrations of Mineral Metabolism

The important roles of several trace minerals, i.e., Mg in the regulation of the antioxidant/pro-oxidant balance within the body through their role in several enzymes and antioxidant enzyme reactions was previously stated [77]. She-camels pre-synchronized using CIDR showed no remarkable differences in serum concentrations of the estimated minerals elements except for P, whereas serum P values were remarkably higher in PREG than those in LACT or in NPREG. In another study, Omidi et al. [28] reported that the concentrations of Ca and P remained unchanged in pregnant she-camels compared to non-pregnant ones. Contrary to the results reported by Omidi et al. [28], Saeed et al. [67] stated that Ca and P concentrations were lower in pregnant she-camels than those in the non-pregnant ones. The current work revealed no remarkable differences in serum Ca, P and Mg concentrations between LACT and NPREG. Serum concentrations of Ca, Mg and P were within their reference ranges reported by Hassan et al. [40]; Abdalmula et al. [67]; Mohamed and Hussein [70]. The similar levels of Ca and P in pregnant and non-pregnant camels might be attributable to a balance between the increase in their absorption from the intestine and sufficient supply these minerals to the fetus [78].

### 4.7. Correlation between Steroid Hormones and Oxidant/Antioxidant Biomarkers

Significant correlations were reported between steroid hormones and oxidant/antioxidant biomarkers. Positive correlations were demonstrated between P_4_ and most antioxidant indicators including TAC, CA and GSH. A negative correlation was reported between P_4_ and SOD. A positive correlation was reported between both E_2_ and cortisol, and SOD. Meanwhile, negative correlations were demonstrated between E_2_ and cortisol, and other antioxidant biomarkers including TAC, CA and GSH. These correlations strongly support our present finding of higher MDA and lower antioxidants in pregnant she-camels. Such status is likely related with the lower immune defense mechanism under high P_4_ levels [60]. Serum P_4_ and cortisol values were negatively correlated with MDA, however, a positive correlation was described between serum E_2_ and serum MDA. Serum P_4_ was negatively correlated with both E_2_ and cortisol, hence a positive correlation was demonstrated between E_2_ and cortisol. On other hand, in she-camels received CIDR, re-used CIDR or OvSynch protocols, negative correlations observed in camels of this study between P_4_ and MDA [10]. In contrast to the negative correlations reported in she-camels of the current study between P_4_ and MDA, cows showed a reduction in oxidative stress indicators together with P_4_ from the time the CIDR [79] and P_4_ release intra-vaginal device were removed [80]. Pregnancy status in she-camels correlated negatively with SOD and positively with alanine aminotransferase. On the other and, lactation status correlated negatively with P_4_ levels, and positively with TAC [23].

### 4.8. Correlation between Steroid Hormones and Liver Functions and Lipid Profile Indices

Positive correlation was found between AST activities and P_4_ concentrations, while no remarkable correlations were reported in she-camels between P_4_ and liver functions/lipid profiles indices except for serum. Regarding to E_2_, serum E_2_ values were negatively correlated with TP, globulins, AST, ALP and GGT. She-camels pre-synchronized using CIDR showed significant correlations between cortisol and liver function biomarkers whereas serum cortisol concentrations were negatively correlated with serum activities of AST, ALP and GGT. On the other hand, Abd El-Hamid et al. [23] reported negative correlation between pregnancy status and ALP. Lactation status correlated positively with albumins and ALP. The change of total proteins, albumin and globulin around calving might be due to the transfer of albumins and γ-globulins and total proteins from blood to the mammary glands [81].

### 4.9. Correlation between Steroid Hormones and Renal Functions Biomarkers

Steroid hormones were significantly correlated with renal function biomarkers in CIDR pre-synchronized she-camels. Serum P_4_ levels were negatively correlated with serum BUN and Cr values. Moreover, both serum E_2_ and cortisol concentrations were correlated positively with Cr and negatively with CK. Omidi et al. [28] reported positive significant correlations between BUN and Cr, BUN and total proteins, BUN and AST, and Cr and albumins in non-pregnant she-camels. These remarkable correlations were not reported in pregnant ones. It was reported that kidney function in camels was less sensitive to dehydration compared to other species [33]. Changes in plasma Cr concentration depended not only on the renal excretion of Cr, but also on its production and volume of distribution. The exceptionally high level of BUN in camels, in comparison to other livestock, was of interest in view of the camel’s ability to utilize urinary nitrogen at times of poor grazing or water deprivation [28].

### 4.10. Correlation between Steroid Hormones and Minerals Parameters

Serum P concentrations were positively correlated with serum concentrations of E_2_ and cortisol, hence, Mg values were negatively correlated with serum P_4_ levels. Kamr et al. [63] reported negative correlation between copper and MDA in pneumonic camels ranging from 2–4 years. Moreover, lower P levels in the non-pregnant she-camels support the possibility that the failure of conception in these two groups would be related with lower P levels [82]. Hassan et al. [40] stated no association was found between low Ca with blood and immune response in diseased camels.

## 5. Conclusions

The present study confirmed the efficacy of using CIDR for synchronization in she-camels. The highest MDA levels were indictors for lipid peroxidation and oxidative stress and the lowest antioxidant levels, i.e., TAC, CAT and GSH, except for SOD in pregnant she-camels, were attributable to physiological oxidative stress. Moreover, lower P levels in non-pregnant she-camels would be contributed to failure of conception or early embryonic death. The investigated she-camels revealed significant correlations between steroid hormones and oxidant indicators, antioxidant biomarkers, lipid profile indices and renal functions biomarkers that provided a better understanding of physiological stress during pregnancy in camels.

## Figures and Tables

**Figure 1 vetsci-08-00247-f001:**
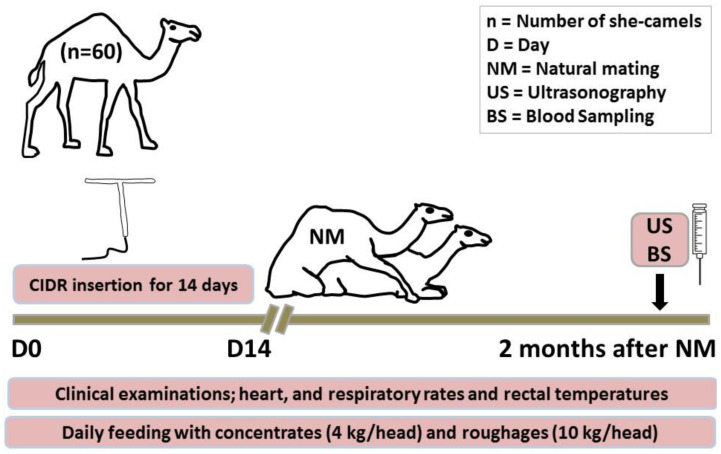
Schematic diagram showing the experimental protocol for the study. She-camels (*n* = 60) were subjected to pre-synchronization using controlled-internal drug releasing (CIDR) for 14 days (D0: day of the insertion of CIDR). After removal of CIDR, pre-synchronized she-camels were subjected to natural mating (NM). In all naturally mated she-camels, ultrasonography (US) and blood sampling (BS) for further biochemical and hormonal assay were performed 2 months after NM. Clinical examinations including assessment of heart and respiratory rates and rectal temperatures were performed. Daily feeding with concentrates (4 kg/head) and roughages (10 kg/head).

**Figure 2 vetsci-08-00247-f002:**
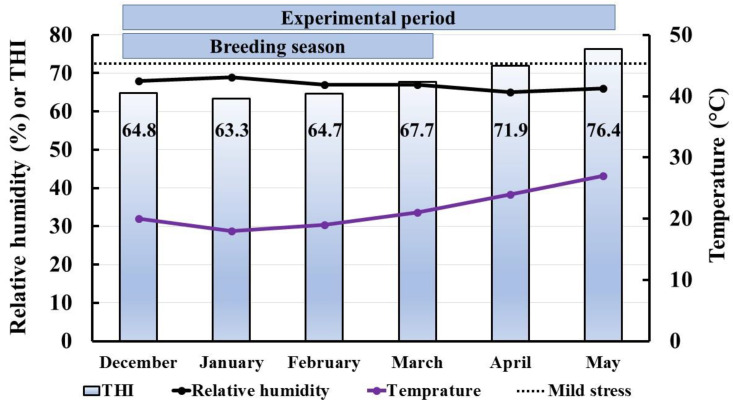
The temperature–humidity index (THI) and the weather data in Alexandria during the experimental period of the study (December to May). The figure is showing the weather data included relative humidity (%; black line), maximum temperature (°C; violet line) and THI (gradient column). The threshold level for mild stress (72–79 THI) is shown as square dot line. The values inside each column refers to the THI value during this month.

**Table 1 vetsci-08-00247-t001:** Mean values (M ± SD) of temperature, pulse, respiration and rumen movements in non-pregnant non-lactating (*n* = 14), non-pregnant lactating (*n* = 8) and pregnant (*n* = 38) she-camels.

	NPREG	LACT	PREG	Reference Values
Temperature (°C)	37.66 ± 0.48 ^a^	37.27 ± 0.32 ^a^	37.82 ± 0.18 ^a^	(37.2 ± 0.77) ^1^ or (37.52 ± 0.09) ^2^
Pulse (Beats/min)	30.56 ± 3.01 ^b^	28.16 ± 2.31 ^b^	36.02 ± 4.15 ^a^	(32–36) ^3^ or (24–48/min) ^4^
Respiration (Breaths/min)	10.66 ± 1.88 ^b^	12.61 ± 3.12 ^b^	20.72 ± 2.03 ^a^	(12.55 ± 0.30) ^2^ or (8–18) ^5^
Rumen motility (Movements/2 min)	3.65 ± 0.66 ^a^	3.15 ± 0.33 ^a^	3.80 ± 0.71 ^a^	(4.25 ± 0.14) ^2^ or (4.3 ± 0.14) ^6^

NPREG: non-pregnant and non-lactating she-camels. LACT: non-pregnant and lactating she-camels. PREG: pregnant she-camels. ^ab^ Means within the same row with different superscript letters in different she-camels group were significantly different. ^1^ Reference values according to Hamad et al. (2017); ^2^ reference values according Hassan et al. (2019); ^3^ reference values according to Fowler et al. (2010); ^4^ reference values according to Bhatt et al. (1960); ^5^ reference values according to Nielsen (1964); ^6^ reference values according to Kamr et al. (2020).

**Table 2 vetsci-08-00247-t002:** Mean values (M ± SD) of P_4_, E_2_ and cortisol in non-pregnant non-lactating (*n* = 14), non-pregnant lactating (*n* = 8) and pregnant (*n* = 38) she-camels.

	NPREG	LACT	PREG	Reference Values
P_4_ (ng/mL)	0.93 ± 0.21 ^c^	2.30 ± 0.32 ^b^	4.29 ± 0.53 ^a^	(0.29 ± 0.26–0.33 ± 0.44) ^1^
E_2_ (pg/mL)	180.38 ± 50.61 ^b^	123.63 ± 20.37 ^c^	265.5 ± 51.91 ^a^	(1.29 ± 1.44–2.66 ± 1.98) ^1^
Cortisol (nmol/L)	42.32 ± 8.96 ^b^	44.63 ± 7.73 ^b^	74.25 ± 15.28 ^a^	(38.17 ± 3.99) ^2^

NPREG: non-pregnant and non-lactating she-camels; LACT: non-pregnant and lactating she-camels; PREG: pregnant she-camels; P_4_: Progesterone; E_2_: Estradiol. ^a–c^ Means within the same row with different superscript letters in different she-camels group were significantly different. ^1^ Reference values according to Ayoub et al. (2003). ^2^ Reference values according to Saeb et al. (2010).

**Table 3 vetsci-08-00247-t003:** Mean values (M ± SD) of MDA, SOD, TAC, CAT and GSH in non-pregnant non-lactating (*n* = 14), non-pregnant lactating (*n* = 8) and pregnant (*n* = 38) she-camels.

	NPREG	LACT	PREG	Reference Values
MDA (μmol/L)	11.35 ± 1.63 ^b^	12.50 ± 1.80 ^b^	28.85 ± 1.72 ^a^	(13.2 ± 0.6) ^1^ or (13.89 ± 0.94) ^2^ or (10·23–11·62) ^3^
SOD (U/mL)	4.09 ± 0.14 ^b^	3.89 ± 0.16 ^b^	4.52 ± 0.17 ^a^	(5.0 ± 0.4) ^1^ or (5·11–6·33) ^3^ or (5.52 ± 0.72) ^4^
TAC (mmol/L)	3.40 ± 0.20 ^a^	3.06 ± 0.14 ^b^	1.79 ± 0.09 ^c^	(1.53 ± 0.48) ^4^ or (0.81 ± 0.02) ^5^
CAT (U/L)	24.46 ± 0.99 ^a^	21.98 ± 1.09 ^b^	17.07 ± 1.17 ^c^	(15.7 ± 0.4) ^1^ or (15.87 ± 0.84) ^2^ or (18.80 ± 0.63) ^4^
GSH (mg/dL)	37.68 ± 2.35 ^a^	31.23 ± 4.80 ^b^	17.29 ± 1.63 ^c^	(8.19 ± 0.53) ^4^

NPREG: non-pregnant and non-lactating she-camels; LACT: non-pregnant and lactating she-camels; PREG: pregnant she-camels; MDA: malondialdehyde; SOD: superoxide dismutase; TAC: total antioxidant capacity; CAT: catalase; GSH: reduced glutathione. ^a–c^ Means within the same row with different superscript letters in different she-camel groups were significantly different. ^1^ Reference values according to El-Bahr and El-Deeb (2016); ^2^ reference values according to Saleh et al. (2009); ^3^ reference values according to El-Deeb and Elmoslemany (2015); ^4^ reference values according to Shoieb et al. (2016); ^5^ reference values according to Kamr et al. (2020).

**Table 4 vetsci-08-00247-t004:** Mean values (M ± SD) of liver functions and lipid profiles indices in non-pregnant non-lactating (*n* = 14), non-pregnant lactating (*n* = 8) and pregnant (*n* = 38) she-camels.

	NPREG	LACT	PREG	Reference Values
Glucose (mmol/L)	9.47 ± 0.82 ^a^	8.76 ± 1.14 ^a^	9.89 ± 1.37 ^a^	(5.01–8.03) ^1^ or (6.36 ± 0.35) ^2^ or (4.88–6.97) ^3^
Total proteins (g/L)	58.66 ± 2.70 ^b^	56.42 ± 2.72 ^a^	55.61 ± 2.99 ^a^	(59.7–106.7) ^1^ or (61.20 ± 4.30) ^2^ or (53–78) ^4^
Albumins (g/L)	27.04 ± 2.85 ^a^	24.58 ± 1.69 ^a^	26.23 ± 2.52 ^a^	(12.2–77.5) ^1^ or (38.30 ± 2.10) ^2^ or (30.80 ± 1.38) ^3^
Globulins (g/L)	34.09 ± 3.74 ^a^	29.68 ± 1.08 ^b^	29.38 ± 1.02 ^b^	(18.42–23.87) ^3^ or (16–29) ^5^
AST (U/L)	106.13 ± 6.73 ^c^	117.88 ± 586 ^b^	126.13 ± 7.12 ^a^	(84.1–161.8) ^1^ or (34–148) ^4^
ALP (U/L)	35.5 ± 7.63 ^c^	52.26 ± 4.50 ^b^	76 ± 8.42 ^a^	(40–176) ^4^ or (41–92) ^5^
GGT (U/L)	16.95 ± 1.58 ^c^	25.25 ± 8.96 ^b^	39.69 ± 2.15 ^a^	(12–28) ^4^ or (7–29) ^5^

NPREG: non-pregnant and non-lactating she-camels; LACT: non-pregnant and lactating she-camels; PREG: pregnant she-camels; AST: aspartate aminotransferase; ALP: alkaline phosphatase; GGT: γ-glutamyl transferase. ^a–c^ Means within the same row with different superscript letters in different she-camel groups were significantly different. ^1^ Reference values according to Islam et al. (2019); ^2^ reference values according to Saeb et al. (2010); ^3^ reference values according to Abdalmula et al. (2019); ^4^ reference values according to Mohamed and Hussein (1999); ^5^ reference values according to Kaneko et al. (2008).

**Table 5 vetsci-08-00247-t005:** Mean values (M ± SD) of, BUN, Cr and CK in non-pregnant non-lactating (*n* = 14), non-pregnant lactating (*n* = 8) and pregnant (*n* = 38) she-camels.

	NPREG	LACT	PREG	Reference Values
BUN (mmol/L)	6.42 ± 0.56 ^b^	9.78 ± 3.26 ^a^	10.60 ± 4.02 ^a^	(13.98 ± 1.09) ^1^ or (2.50–9.64) ^2^
Cr (μmol/L)	72.29 ± 9.15 ^b^	76.80 ± 13.85 ^b^	93.93 ± 10.09 ^a^	(114.92–265.20) ^2^ or (97.2–221) ^3^
CK (U/L)	105.99 ± 5.81 ^b^	107.88 ± 4.39 ^b^	113.38 ± 4.44 ^a^	(70–250) ^2^ or (100.36–159·76) ^4^

NPREG: non-pregnant and non-lactating she-camels; LACT: non-pregnant and lactating she-camels; PREG: pregnant she-camels; BUN: blood urea nitrogen; Cr: creatinine; CK: creatine kinase. ^ab^ Means within the same row with different superscript letters in different she-camel groups were significantly different. ^1^ Reference values according to Saeb et al. (2010); ^2^ reference values according to Mohamed and Hussein (1999); ^3^ reference values according to Kaneko et al. (2008); ^4^ reference values according to El-Deeb and Elmoslemany (2015).

**Table 6 vetsci-08-00247-t006:** Mean values (M ± SD) of Ca, P and Mg in non-pregnant non-lactating (*n* = 14), non-pregnant lactating (*n* = 8) and pregnant (*n* = 38) she-camels.

	NPREG	LACT	PREG	Reference Values
Ca (mmol/L)	2.28 ± 0.38 ^a^	2.21 ± 0.09 ^a^	2.08 ± 0.21 ^a^	(1.80–3.13) ^1^ or (1.43–2.12) ^2^ or (2.63 ± 0.07) ^3^
P (mmol/L)	1.54 ± 0.30 ^b^	1.57 ± 0.26 ^b^	1.92 ± 0.25 ^a^	(1.10–2.29) ^1^ or (1.43–2.12) ^2^ or (3.53 ± 0.08) ^3^
Mg (mmol/L)	0.94 ± 0.08 ^a^	1.09 ± 0.11 ^a^	1.02 ± 0.09 ^a^	(0.74–1.72) ^1^ or (1.07–1.01) ^2^ or (1.07 ± 0.03) ^3^

NPREG: non-pregnant and non-lactating she-camels; LACT: non-pregnant and lactating she-camels; PREG: pregnant she-camels; Ca: calcium; P: phosphorous; Mg: magnesium. ^a,b^ Means within the same row with different superscript letters in different she-camel groups were significantly different. ^1^ Reference values according to Mohamed and Hussein (1999); ^2^ reference values according to Abdalmula et al. (2019); ^3^ reference values according to Hassan et al. (2019).

**Table 7 vetsci-08-00247-t007:** Pearson correlation coefficient between steroids hormones, MDA, SOD, TAC, CAT and GSH in non-pregnant non-lactating (*n* = 14), non-pregnant lactating (*n* = 8) and pregnant (*n* = 38) she-camels.

	P_4_ (ng/mL)	E_2_ (pg/mL)	Cortisol (nmol/L)	MDA (μmol/L)	SOD (U/mL)	TAC (mmol/L)	CAT (U/L)	GSH (mg/dL)
**P_4_**		−0.1019	−0.4394 **	−0.5116 **	−0.6291 **	0.2936	0.1703	0.1908
		(0.6358)	(0.0317)	(0.0106)	(0.0010)	(0.1637)	(0.4263)	(0.3718)
**E_2_**			0.5821 **	0.7590 **	0.57059 **	−0.7703 **	−0.7371 **	−0.8106 **
			(0.0028)	(0.0002)	(0.00360)	(0.0001)	(0.0004)	(0.0002)
**Cortisol**				−0.8060 **	0.6514 **	−0.7663 **	−0.7499 **	−0.7398 **
				(0.0001)	(0.0006)	(0.0001)	(0.0002)	(0.0004)
**MDA**					0.83926 **	−0.9201 **	−0.8534 **	−0.8624 **
					(0.0003)	(0.0002)	(0.0001)	(0.0006)
**SOD**						−0.7371 **	−0.6095 **	−0.6855 **
						(0.0004)	(0.0016)	(0.0002)
**TAC**							0.9198 **	0.8873 **
							(0.0002)	(0.0008)
**CAT**								0.8674 **
								(0.0004)
**GSH**								


P_4_: Progesterone; E_2_: Estradiol; MDA: malondialdehyde; SOD: superoxide dismutase; TAC: total antioxidant capacity; CAT: catalase; GSH: reduced glutathione; R = Correlation. ** Significant (two-tailed) *p* < 0.01. Gray backgrounds referred to correlation between the same parameter e.g., P_4_ and P_4_ where there was no correlation. Diagonal backgrounds referred that this correlation was previously reported in the previous row e.g., correlation between P_4_ and E_2_ was reported in row 1 and so there was no need to repeat it at row 2.

**Table 8 vetsci-08-00247-t008:** Pearson correlation coefficient between steroid hormones, glucose, total proteins, albumins, globulins, AST, ALP and GGT in non-pregnant non-lactating (*n* = 14), non-pregnant lactating (*n* = 8) and pregnant (*n* = 38) she-camels.

	P_4_ (ng/mL)	E_2_ (pg/mL)	Cortisol (nmol/L)	Glucose (mmol/L)	TP (g/L)	Albumins (g/L)	Globulins (g/L)	AST (U/L)	ALP (U/L)	GGT (U/L)
**P_4_**				0.0723	−0.1012	0.3344	−0.2784	0.5081 *	−0.0885	−0.03903
				(0.1878)	(0.6379)	(0.1102)	(0.1878)	(0.0113)	(0.6810)	(0.8563)
**E_2_**				0.1472	−0.4968 *	0.2550	−0.6352 **	−0.3715 *	−0.6997 **	−0.8219 **
				(0.4926)	(0.0135)	(0.2291)	(0.0085)	(0.0439)	(0.0001)	(0.0008)
**Cortisol**				0.2241	−0.1862	0.3381	−0.2665	−0.4431 *	−0.5923 **	−0.6433 **
				(0.2926)	(0.3836)	(0.1061)	(0.2081)	(0.0301)	(0.0023)	(0.0070)
**Glucose**					−0.1618	0.0338	−0.0603	−0.3569	−0.3613	−0.3447
					(0.4410)	(0.8753)	(0.7795)	(0.0869)	(0.0828)	(0.0990)
**TP**						0.4298 *	0.4437 *	−0.4315 *	0.5015 *	−0.3569
						(0.0361)	(0.0299)	(0.0352)	(0.0125)	(0.0869)
**Albumins**							−0.5276 **	0.1573	−0.2825	−0.3332
							(0.0080)	(0.4628)	(0.1811)	(0.1117)
**Globulins**								−0.1116	0.3988	0.5712 **
								(0.6037)	(0.0536)	(0.0036)
**AST**									0.4298 *	0.4437 *
									(0.0361)	(0.0299)
**ALP**										0.9009 **
										(0.0002)
**GGT**										


P_4_: Progesterone; E_2_: Estradiol; TP: total proteins; AST: aspartate aminotransferase; ALP: alkaline phosphatase; GGT: γ-glutamyl transferase. * Significant (two-tailed) *p* < 0.05. ** Significant (two-tailed) *p* < 0.01. Gray backgrounds referred to correlation between the same parameter e.g., P_4_ and P_4_ where there was no correlation. Diagonal backgrounds referred that this correlation was previously reported either in Table 7 or in the previous row e.g., correlation between P_4_ and E_2_ was reported in Table 7 and so there was no need to repeat it in Table 8.

**Table 9 vetsci-08-00247-t009:** Pearson correlation coefficient between steroids hormones, BUN, Cr and CK in non-pregnant non-lactating (*n* = 14), non-pregnant lactating (*n* = 8) and pregnant (*n* = 38) she-camels.

	P_4_ (ng/mL)	E_2_ (pg/mL)	Cortisol (nmol/L)	BUN (mmol/L)	Cr (μmol/L)	CK (U/L)
**P_4_**				−0.5141 *	−0.4100 *	−0.1148
				(0.0102)	(0.0466)	(0.5934)
**E_2_**				0.0181	0.5278 **	−0.4815 *
				(0.9331)	(0.0080)	(0.0172)
**Cortisol**				0.0129	0.5921 **	−0.2811
				(0.9525)	(0.0023)	(0.1833)
**BUN**					0.1295	−0.0104
					(0.5464)	(0.9617)
**Cr**						−0.3594
						(0.0845)
**CK**						


P_4_: Progesterone; E_2_: Estradiol; BUN: blood urea nitrogen; Cr: creatinine; CK: creatine kinase. * Significant (two-tailed) *p* < 0.05. ** Significant (two-tailed) *p* < 0.01. Gray backgrounds referred to correlation between the same parameter e.g., P4 and P4 where there was no correlation. Diagonal backgrounds referred that this correlation was previously reported either in Table 8 or in the previous row e.g., correlation between P_4_ and E_2_ was reported in Table 8 and so there was no need to repeat it in Table 9.

**Table 10 vetsci-08-00247-t010:** Pearson correlation coefficient between steroids hormones, Ca, P and Mg in non-pregnant non-lactating (*n* = 14), non-pregnant lactating (*n* = 8) and pregnant (*n* = 38) she-camels.

	P_4_ (ng/mL)	E_2_ (pg/mL)	Cortisol (nmol/L)	Ca (mmol/L)	P (mmol/L)	Mg (mmol/L)
**P_4_**				0.3342	−0.2719	−0.4823 *
				(0.110412)	(0.1987)	(0.0170)
**E_2_**				−0.1825	0.5805 **	−0.1231
				(0.39350)	(0.0029)	(0.5665)
**Cortisol**				−0.3060	0.4165 *	−0.0446
				(0.1459)	(0.0429)	(0.8362)
**Ca**					0.0618	0.2237
					(0.7741)	(0.2935)
**P**						0.4603 *
						(0.0236)
**Mg**						


P_4_: Progesterone; E_2_: Estradiol; Ca: calcium; P: phosphorous; Mg: magnesium. * Significant (two-tailed) *p* < 0.05. ** Significant (two-tailed) *p* < 0.01. Gray backgrounds referred to correlation between the same parameter e.g., P_4_ and P_4_ where there was no correlation. Diagonal backgrounds referred that this correlation was previously reported either in Table 9 or in the previous row e.g., correlation between P_4_ and E_2_ was reported in Table 9 and so there was no need to repeat it in Table 10.

## Data Availability

Data is available after publishing in this journal.

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
