# Peer review of "Clinical and Correlated Responses among Steroid Hormones and Oxidant/Antioxidant Biomarkers in Pregnant, Non-Pregnant and Lactating CIDR-Pre-Synchronized Dromedaries (Camelus dromedarius)"

_vetsci, 2021, doi:10.3390/vetsci8110247_

Round 1

Reviewer 1 Report

VETERINARY SCIENCES - MDPI: Referee’s Evaluation Report

MANUSCRIPT IDENTIFICATION:  vetsci-1358434

Clinical findings and correlations of steroid hormones and oxidants/antioxidants biomarkers in non-pregnant, lactating and pregnant one-humped camels (Camelus dromedarius) pre-synchronized by controlled-internal drug releasing. (ORIGINAL ARTICLE)

Comments to Authors/Editor:

The paper of Ragab Mohamaed & colleagues aimed to assess possible changes in steroid hormones, oxidant-antioxidants biomarkers, lipid profiles/liver functions indices, renal function biomarkers & mineral metabolism indicators in pregnant, non-pregnant, & lactating one-humped camels (Camelus dromedarius) pre-synchronized by controlled-internal drug releasing.  This manuscript falls within the scope of Veterinary Sciences. The manuscript is sufficiently informative for the replication of the study.  In general, the organization of the experiment seems to be well designed, yet, the English quality, grammar, and sentence structure must be greatly improved. Please, adjust the title; in its actual format, it is quite long. Besides, as I understand, dromedaries and camels are both camelids, yet, while dromedaries have one hump, camels have two; please correct accordingly. The authors may consider this title: “Clinical and correlated responses among steroid hormones and oxidants/antioxidants biomarkers in pregnant, non-pregnant, and lactating dromedaries (Camelus dromedarius). The Abstract is extremely long and was written in a careless fashion. The authors must re-write all this section. Besides, both in the Abstract and M&M sections, the authors must include where the experiment was carried out (NL, WL), including the months of the breeding season and the main environmental indicators (i.e. temperature, humidity, radiation, photoperiod, etc.) along with the experimental period. The authors must include such information especially because this research included some physiological-related variables. Please define if the experimental samples were collected at the beginning, the middle or the last stage of the breeding season.  These issues must be perfectly and carefully addressed. While the authors mentioned that an experienced male was used, they never stated if the male was “sexually active” and if both semen quality and libido tests were performed prior to the experimental period. The experimental groups were: pregnant, non-pregnant & lactating; please simplify the acronyms: PREG, NPREG & LACT, respectively.  The results part of the abstract is particularly confusing; this information must be rewritten. Also, the authors must simplify the information because of the large set of response variables, and only include those response variables which differed among experimental groups. The authors must select some key variables to include the observed value at least in the best experimental group along with the probability level. Why did the authors finish the abstract addressing the concept of “physiological stress” (i.e. “the negative approach”), instead of the excellent compensatory responses observed in the PREG group to face such physiologic stage (i.e. “the positive approach”)???  It is important to include the temperature-humidity index (THI) observed along with the experimental period, to better relate the physiologic responses of females to their environmental context. The authors must follow this strategy along with the whole manuscript. No Simple Summary was included. In the Introduction section, every statement must include at least one reference; correct accordingly along with the manuscript. L75 & 77, “were” or “are”??? L79, “poor reproductive performance”, or “wise-strategic reproductive performance”; do these animals have the best environmental condition (i.e. temperature, humidity, available feeding, and supplementation), to expect an “excellent reproductive performance”???; are the authors comfortable with this approach?? L83, low breeding season??, what do you mean by that? The “scientific structure” used by the authors in the introduction section is not the most logical and sensible format. The Introduction section is quite long; are the authors comfortable with 85 lines as an Introduction??; this section must be shortened. Some information must be moved to the Discussion section. While no working hypothesis of the study was proposed, the objectives of the study were clearly stated.  Please include the national dromedary inventory and the contribution of dromedaries to the national livestock sector; why the authors used female dromedaries instead of female camels??? Are camelids, and more specifically dromedaries, important from a productive, economic, or social standpoint??? In the Material & Methods section, I do strongly recommend moving section 2.2. as section 2.1. Then, section 2.2 could involve information regarding “Location, environmental conditions, and experimental groups”, 2.3. Reproductive management. As mentioned, it is recommended to include information regarding the location, the main prevailing environmental conditions of temperature, humidity, radiation, photoperiod along with the experimental period. I strongly recommend including a figure with the actual experimental protocol across time (i.e. a timeline of actions); this is a must. Besides, the authors must indicate how the females were managed during the experimental period. Do the animals were managed as treatment groups in the same or different corrals?? What was the area (i.e. m2 per animal) for each treatment group? Were the animals group-feed or individually-feed??  Please clarify. Reagents, standards, and methods used are relevant and in accordance with the objectives of the study. Also, all the treatment, sampling techniques, laboratory methods, as well as response variables considered in the experiment are detailed and accurate, while in agreement with the objectives of the study.  L192, “sera” or “serum”?? L199, Hormonal analysis or analyses???, what methodology was used for hormone quantification, ELISA, RIA, or other??? Please clarify and include the required information regarding the standards and CV´s, or other required information. Besides, 2.7. liver functions or Liver function??, 2.8, kidney functions or Kidney function??, 2.10, Statistical analysis or Statistical analyses??? The experimental design, and statistical models, were described well enough for the reader to understand how the experiment was carried out.  However, the authors must explain why they are reporting means instead of least-square means? Also, describe if the response variables depicted a normal distribution or if they required an adjustment or transformation in order to perform the analyses through ANOVA. Are the Pearson Correlation analyses accurate for all the evaluated variables??; why the authors decided not to use the Spearman Correlation procedure for some defined variables?? Regarding the Results section, the novelty value of the results is reasonable. In L238, and along with the Discussion section, the authors use the term “no remarkable differences”; do the authors feel comfortable with the use of such word??, better use significant differences; is not between, is among groups.  Besides, the authors must include some kind of quantitative information aside from the P-values of the observed results regarding the response variables.  Also, if no differences occurred among experimental groups for a defined variable, the authors must include the average value for such response variables observed in the study. Further, the inclusion of the “p-values” along with the word “significantly” is a pleonasm; just use one, not both; correct accordingly along with the whole manuscript as well as Tables; this is a must. Results were shown in 10 Tables; the titles must be rewritten; the authors must include the number of replicates per experimental group, and mention if they collected repeated samples across time within the experimental group. The titles of tables must stand by themselves; titles must be rewritten.  L294, “remarkable increased”????; your statistical analyses among experimental groups were performed to detect any possible DIFFERENCE among groups; please use such approach instead of the “remarkable or significant” approach. Besides, regarding the last tables (i.e. the correlation tables), please eliminate the suffix (r=xxxx), and the word “P-value”; it makes no sense at all to repeat such information along with all the correlated response variables, just include in the superior part the correlation value and above the P-value in parenthesis (i.e. 0.045); the use of such repeated word & suffixes only creates visual contamination. Regarding the Discussion section, at the beginning of the Discussion, I do strongly suggest initiating this section including the working hypothesis of the study. Authors must define if, with the obtained results, such a hypothesis is rejected or non-rejected. For this reason, the authors must include the working hypothesis prior to the objectives in the Introduction section. In addition, the authors must follow the same order in this section according to that proposed in the Results section; the authors must homogenate the presentation of the Results and Discussion.  The authors must link, in a logical fashion, their main findings along with the discussion section, to compare & discuss and, thereafter, be able to propose some physiologic explanations for such specific outcomes, considering previous similar studies from the scientific literature. In general, the authors made an accurate interpretation of the main findings; the authors are using different sizes regarding the type of words in the Discussion.  The Discussion section is quite extended; the authors must focus their main findings and confront them with respect to the scientific literature in a logical and focused fashion.  The authors repeat many ideas already presented in the Results section; this makes so extended this section. The main outcomes of the study were not soundly presented.  Some information is repetitive and must be eliminated; the Discussion section must be shortened and focused in a logical and scientific fashion. The Discussion must be precise, honest, fair, and in accordance with the obtained results and the main objectives of the study. The list of references cited in the manuscript is proper, yet, is quite extended; I strongly suggest eliminating all the bibliography older than the year 2000, also, 60% of the citations must be within the years 2015-2021; this is a must. This is an interesting study. Yet, the authors must improve both the English language quality as well as the clarity and logical arrangement of the observed results, especially in the Discussion section and the Conclusion section; not always more is better.  All the commented issues and requests should be clearly addressed by the authors; at this point, and based on the above comments, my pronouncement is that this manuscript cannot be accepted in its actual format.  It requires extensive editions and corrections.

Author Response

Reviewer: 1]

Comments to the Authors/Editor

The paper of Ragab Mohamaed & colleagues aimed to assess possible changes in steroid hormones, oxidant-antioxidants biomarkers, lipid profiles/liver functions indices, renal function biomarkers & mineral metabolism indicators in pregnant, non-pregnant, & lactating one-humped camels (Camelus dromedarius) pre-synchronized by controlled-internal drug releasing.  This manuscript falls within the scope of Veterinary Sciences. The manuscript is sufficiently informative for the replication of the study.

To Reviewer: 1

We are most grateful to Reviewer 1 for the critical comments and useful suggestions that have helped us to improve our paper considerably.  As indicated in the responses that follow, we have taken all these comments and suggestions into account in the revised version of our paper.  The changes to the text have been indicated by the track changes mode in the revised manuscript.  We sincerely hope the revised manuscript meets your comments.

Your comment:

  1. In general, the organization of the experiment seems to be well designed, yet, the English quality, grammar, and sentence structure must be greatly improved.

Our response:

Thank you for your comment.  As you pointed out in your comment, we revised the manuscript to meet the English quality, grammar, and sentence structure.

----------------------------------------------------------------------

Your comment:

Please, adjust the title; in its actual format, it is quite long. Besides, as I understand, dromedaries and camels are both camelids, yet, while dromedaries have one hump, camels have two; please correct accordingly.

The authors may consider this title: “Clinical and correlated responses among steroid hormones and oxidants/antioxidants biomarkers in pregnant, non-pregnant, and lactating dromedaries (Camelus dromedarius).

Our response:

Thank you for your comment.  We would like to explain that there are two domesticated species of camels; one-humped camels known as Dromedary camels, and two-humped camels known as Bactrian camels. In our study, we used the one-humped dromedary camels.

Regarding the title, Thank you so much for your comment. We changed the title into “Clinical and correlated responses among steroid hormones and oxidants/antioxidants biomarkers in pregnant, non-pregnant, and lactating CIDR-pre-synchronized dromedaries (Camelus dromedarius)”.

----------------------------------------------------------------------

Your comment:

The Abstract is extremely long and was written in a careless fashion. The authors must re-write all this section. The results part of the abstract is particularly confusing; this information must be rewritten.

Our response:

Thank you for your comment.  We have revised the Abstract as recommended. We hope that our revisions meet Reviewer 1 recommendations.

----------------------------------------------------------------------

Your comment:

Besides, both in the Abstract and M&M sections, the authors must include where the experiment was carried out (NL, WL), including the months of the breeding season and the main environmental indicators (i.e. temperature, humidity, radiation, photoperiod, etc.) along with the experimental period. The authors must include such information especially because this research included some physiological-related variables.

Our response:

Thank you for your comment.  We would like to explain that we added the required details about the study location where the experiment was conducted as follow:

Study location

Dromedary she-camels used in this study reared at Mariut Research Station, Desert Research Center, El-Amria, Alexandria, Egypt.

Alexandria governorate, Egypt where the study was conducted, located at -1 m above mean sea level, latitude 31.21° N and longitude 29.95° E. The study was performed during the period from December to May when the low (minimum) and high (maximum, °C) temperature ranged between 10-17°C and 20-27°C, respectively. The relative humidity ranged between 66% in May and 68% in December. The photoperiod throughout the whole period of the study extended from 10:15 hrs in December to 12:30 hrs in May.

----------------------------------------------------------------------

Your comment:

The authors must include such information especially because this research included some physiological-related variables. Please define if the experimental samples were collected at the beginning, the middle or the last stage of the breeding season.  These issues must be perfectly and carefully addressed

Our response:

Thank you for your comment.  We have revised the Materials and Methods section as recommended. We hope that our revisions meet your valuable recommendations.

Kindly find the revised part as follow;

2.4. Samples

Blood samples were collected from the jugular vein at same time-point of ultrasonography (at the end of the breeding season, 2-months after the natural mating) into vacutainer tubes with sodium (Na) fluoride for determination of glucose levels in plasma and plain vacutainer tubes for separation of sera and measuring different biochemical parameters. Serum and plasma samples were collected and kept frozen at -20 °C for subsequent hormonal and biochemical analyses using commercial test kits according to the standard protocols of suppliers.

----------------------------------------------------------------------

Your comment:

While the authors mentioned that an experienced male was used, they never stated if the male was “sexually active” and if both semen quality and libido tests were performed prior to the experimental period.

Our response:

Thank you for your comment.  We used sexually active camel-bulls of good fertility. The breeding history of the selected fertile camel-bulls was checked. And the sexual desire of the used fertile camel-bulls was evaluated by showing rutting behaviour when introduced to she-camel(s) in heat. This part has been revised and explained as recommended.

Kindly find the revised part as follow:

An experienced sexually active camel-bull of good fertility (with good breeding history) was introduced to she-camels after the removal of CIDR. The sexual desire of the used fertile camel-bulls was examined by showing rutting behaviour when introduced to she-camel(s) in heat.

Instead of “An experienced male was introduced to she-camels after the removal of CIDR”.

----------------------------------------------------------------------

Your comment:

I strongly recommend including a figure with the actual experimental protocol across time (i.e. a timeline of actions); this is a must.

Our response:

Thank you for your comment.  We added a schematic diagram for the actual experimental protocol (Figure 1) as recommended. We hope this figure meets Reviewer 1 recommendation.

The figure legend as follow:

Figure 1. Schematic diagram showing the experimental protocol for the study. She-camels (n=60) were subjected to pre-synchronization using controlled-internal drug releasing (CIDR) for 14 days (D0: day of the insertion of CIDR). Three to 10 days after removal of CIDR (D17-D24), pre-synchronized she-camels showing heat were subjected to natural mating (NM). In all naturally mated she-camels, ultrasonography (US) and blood sampling (BS) for further biochemical and hormonal assay were performed on D77-D84 (2-months after NM). Clinical examinations including assessment of heart, and respiratory rates and rectal temperatures were performed. Daily feeding with concentrates (4 kg/head) and roughages (10 kg/head).

----------------------------------------------------------------------

Your comment:

Besides, the authors must indicate how the females were managed during the experimental period. Do the animals were managed as treatment groups in the same or different corrals??

Our response:

Thank you for your comment.  We would like to explain that all animals were reared in the same corrals; in an open yard under the same circumstances. More details about the management were explained in Figure 1 that we added in the revised manuscript.

----------------------------------------------------------------------

Your comment:

What was the area (i.e. m2 per animal) for each treatment group?

Our response:

Thank you for your comment.  We would like to explain that all camels were housed in an open yard of 1300 m2 area. Each head has a space allowance of more than 20 m2. We hope that our revisions meet Reviewer 1 recommendations.2

Reviewer 2 Report

The authors submit a very articulated manuscript on the physiological status in pregnant she-one-humped camels after CIDR treatment, emphasizing the antioxidant aspetta. The manuscript is in  in my opinion, very well written, articulated and complete. The introduction section is in my opinion a little dispersive, but clear. The methods appear adequate and well-described, and the statistics appropriate.

Therefore I recommend the acceptance of the manuscript for publication on Veterinary Sciences, in the present form.

Author Response

 [Reviewer: 2]

Comments to the Authors/Editor

The authors submit a very articulated manuscript on the physiological status in pregnant she-one-humped camels after CIDR treatment, emphasizing the antioxidant aspetta. The manuscript is in   in my opinion, very well written, articulated and complete. The introduction section is in my opinion a little dispersive, but clear. The methods appear adequate and well-described, and the statistics appropriate.

Therefore, I recommend the acceptance of the manuscript for publication on Veterinary Sciences, in the present form.

To Reviewer: 2

We are most grateful to Reviewer 2 for the critical comments and useful suggestions that have helped us to improve our paper considerably.  

Reviewer 3 Report

This is an interesting study related to the estrus synchronization and to the clinical findings and correlations of steroid hormones and oxidants/antioxidants biomarkers in  one-humped camels. At general, manuscript is well written, but some minor points should be revised before final acceptance. The main considerations are listed below:

  1. Tittle - it sounds adequate.

2. Abstract - It seeems to be very long, but this is not a problem since it is into the limit of characters. However, authors should present only the main findings, also providing some numeric results related to the means found for hormones or biomarkers. They should also indicate the main correlations found by expressing the values for these correlations. Conclusions should be more objective. 

3. Keywords - Please revise the word "indicator" both in abstract as in keywords.

4. Introduction - It is well written and clearly demonstrate statements that justify the execution of the study. However, it is so extensive and brings a lot of unnecessary literature review. Authors are advised to reduce the introduction. In my point of view, they should reduce information related to oxidative stress (from line 92 to 130). Moreover, please revise a reference (Monaco et al., 2013) in line 89. The last paragraph related to objectives is so detailed and some of these details could only be presented at the methodology section. Please revise it.

5. Materials and Methods

  • Since camels are seasonal breeders as reported at the introduction, authors should provide the geographical coordinates where the study was conducted, also pointing the photoperiod to which animals were subjected during experiment.
  • Ethical guidelines should be the first topic of the section.
  • It is not clear when the bllood samples were collected during experiments. Were they collected and checked only once? Please detail the calendar for blood collections. The same for clinical examinations, where authors should provide information related to the moment of the day when examinations were conducted, since environmental variables could interfere in some parameters.
  • Authors should provide literature references for methodologies as hormone analysis, oxidant and antioxidant biomarkers assays, liver functions and lipid profile indexes, etc

6. Results

  • If the first objective of the study was to check the efficiency of estrous synchronization using CIDR, authors should provide this results first. I understand that this was essential for the definition of experimental groups, but it is also a result and should be highlighted here. Auhors should provide percentual values related to the rate of individuals that efficiently responded to the synch protocol.
  • At general, results are well presented. Research is very rich in details, and this sounds really good. Even if a lot of tables is presented, all of them seems to be essential for the best presentation of results.

7. Discussion, conclusions and references

  • I am particularly satisfied at seeing discussions separated in different sections. This help us to a better understanding of the statements.
  • At general, all the main points are well discussed.
  • Conclusions are objective and adequate.
  • References are recent and adequate.

Author Response

[Reviewer: 3]

Comments to the Authors/Editor

This is an interesting study related to the estrus synchronization and to the clinical findings and correlations of steroid hormones and oxidants/antioxidants biomarkers in one-humped camels. At general, manuscript is well written, but some minor points should be revised before final acceptance. The main considerations are listed below:

To Reviewer: 3

We are most grateful to Reviewer 3 for the critical comments and useful suggestions that have helped us to improve our paper considerably.  As indicated in the responses that follow, we have taken all these comments and suggestions into account in the revised version of our paper.  The changes to the text have been indicated by the track changes mode in the revised manuscript.  We sincerely hope the revised manuscript meets your comments.

Your comment:

  1. Tittle - it sounds adequate.

Our response:

Thank you for your comment.  

----------------------------------------------------------------------

Your comment:

  1. Abstract - It seems to be very long, but this is not a problem since it is into the limit of characters. However, authors should present only the main findings, also providing some numeric results related to the means found for hormones or biomarkers. They should also indicate the main correlations found by expressing the values for these correlations. Conclusions should be more objective.

Our response:

Thank you for your comment.  We have revised the Abstract as recommended. We hope that our revisions meet Reviewer 2 recommendations.

----------------------------------------------------------------------

Your comment:

  1. Keywords - Please revise the word "indicator" both in abstract as in keywords.

Our response:

Thank you for your comment.  We revised it:

----------------------------------------------------------------------

Your comment:

  1. Introduction - It is well written and clearly demonstrate statements that justify the execution of the study. However, it is so extensive and brings a lot of unnecessary literature review. Authors are advised to reduce the introduction. In my point of view, they should reduce information related to oxidative stress (from line 92 to 130). Moreover, please revise a reference (Monaco et al., 2013) in line 89. The last paragraph related to objectives is so detailed and some of these details could only be presented at the methodology section. Please revise it.

Our response:

Thank you for your comment.  We have revised introduction section as recommended. We hope that our revisions meet your valuable recommendations.

----------------------------------------------------------------------

Your comment:

Since camels are seasonal breeders as reported at the introduction, authors should provide the geographical coordinates where the study was conducted, also pointing the photo period to which animals were subjected during experiment.

Ethical guidelines should be the first topic of the section.

It is not clear when the blood samples were collected during experiments. Were they collected and checked only once? Please detail the calendar for blood collections. The same for clinical examinations, where authors should provide information related to the moment of the day when examinations were conducted, since environmental variables could interfere in some parameters.

Authors should provide literature references for methodologies as hormone analysis, oxidant and antioxidant biomarkers assays, liver functions and lipid profile indexes, etc.

Our response:

Study location

Dromedary she-camels used in this study reared at Mariut Research Station, Desert Research Center, El-Amria, Alexandria, Egypt.

Alexandria governorate, Egypt where the study was conducted, located at -1 m above mean sea level, latitude 31.21° N and longitude 29.95° E. The study was performed during the period from December to May when the low (minimum) and high (maximum, °C) temperature ranged between 10-17°C and 20-27°C, respectively. The relative humidity ranged between 66% in May and 68% in December. The photoperiod throughout the whole period of the study extended from 10:15 hrs in December to 12:30 hrs in May.

We have revised the Materials and Methods section as recommended. We hope that our revisions meet your valuable recommendations.

Kindly find the revised part as follow;

2.4. Samples

Blood samples were collected from the jugular vein at same time-point of ultrasonography (at the end of the breeding season, 2-months after the natural mating) into vacutainer tubes with sodium (Na) fluoride for determination of glucose levels in plasma and plain vacutainer tubes for separation of sera and measuring different biochemical parameters. Serum and plasma samples were collected and kept frozen at -20 °C for subsequent hormonal and biochemical analyses using commercial test kits according to the standard protocols of suppliers.

We would like to explain that all camels were housed in an open yard of 1300 m2 area. Each head has a space allowance of more than 20 m2.

----------------------------------------------------------------------

Your comment:

  1. Results

If the first objective of the study was to check the efficiency of estrous synchronization using CIDR, authors should provide this results first. I understand that this was essential for the definition of experimental groups, but it is also a result and should be highlighted here. Auhors should provide percentual values related to the rate of individuals that efficiently responded to the synch protocol.

At general, results are well presented. Research is very rich in details, and this sounds really good. Even if a lot of tables is presented, all of them seems to be essential for the best presentation of results.

Our response:

Thank you for your comment.  We added the percentual values to the tables.

----------------------------------------------------------------------

Your comment:

Besides, the authors must indicate how the females were managed during the experimental period. Do the animals were managed as treatment groups in the same or different corrals??

Our response:

Thank you for your comment.  We would like to explain that all animals were reared in the same corrals; in an open yard under the same circumstances. More details about the management were explained in Figure 1 that we added in the revised manuscript.

----------------------------------------------------------------------

Your comment:

  1. Discussion, conclusions and references

I am particularly satisfied at seeing discussions separated indifferent sections. This help us to a better understanding of the statements.

At general, all the main points are well discussed.

Conclusions are objective and adequate.

References are recent and adequate.

Our response:

Thank you for your comment. 

Round 2

Reviewer 1 Report

VETERINARY SCIENCES - MDPI: Referee’s Evaluation Report

MANUSCRIPT IDENTIFICATION:  vetsci-1358434-R1

Clinical findings and correlated responses among steroid hormones and oxidants/antioxidants biomarkers in pregnant, non-pregnant, and lactating CDIR-pre-synchronized dromedaries (Camelus dromedarius) (ORIGINAL ARTICLE)

Comments to Authors/Editor:

The paper of Ragab Mohamed & colleagues aimed to assess possible changes in steroid hormones, oxidant-antioxidants biomarkers, lipid profiles/liver functions indices, renal function biomarkers & mineral metabolism indicators in pregnant, non-pregnant, & lactating one-humped camels (Camelus dromedarius) pre-synchronized by controlled-internal drug releasing.  This manuscript falls within the scope of Veterinary Sciences. This new R1 version of the manuscript has been certainly improved. The English quality, grammar, and sentence structure were certainly amended, but the manuscript requires some other correction-adjustments. Both the title and the Abstract were adjusted. While the authors mentioned that an experienced male was used, this new R1 version confirms that they used male was “sexually active”. Besides, the authors simplified the identification of the treatment groups with the use of the acronyms: PREG, NPREG & LACT do recommend.  The results section of the abstract is very long and particularly confusing; I do recommend including in one “block” all the positive correlations, and then, all the negative correlations, but only those denoting significant correlations, either positive or negative. As previously suggested, the authors must select some key variables to include the observed value at least in the best experimental group along with the probability level. In this new R1 version, the authors use the “positive approach” regarding the excellent compensatory responses observed in the PREG group to face such a physiologic stage.  The abstract sections MUST be shortened. It was previously mentioned that is important to include the temperature-humidity index (THI) observed along with the experimental period, to better relate the physiologic responses of females to their environmental context. No Simple Summary was included. In the Introduction section, every statement included at least one reference; this section is quite long; it must be shortened. Some information must be moved to the Discussion section. Again, no working hypothesis of the study was proposed; this is a must. Besides, as suggested, include the national dromedary inventory, and the contribution of dromedaries to the national livestock sector. The Material & Methods section was certainly improved. Reagents, standards, and methods used are relevant and in accordance with the objectives of the study. Also, all the treatment, sampling techniques, laboratory methods, as well as response variables considered in the experiment are detailed and accurate, while in agreement with the objectives of the study.  The experimental design, and statistical models, were described well enough for the reader to understand how the experiment was carried out.  However, whereas the authors did not explain why they are reporting means instead of least-square means, they also never mentioned why they used Pearson´s Correlation Analyses instead of the Spearman Correlation procedure for some defined variables. Regarding the Results section, the novelty value of the results is reasonable. The authors must include some kind of quantitative information aside from the P-values of the observed results regarding the response variables.  Also, if no differences occurred among experimental groups for a defined variable, the authors must include the average value for such response variables observed in the study. Further, as previously mentioned, the inclusion of the “p-values” along with the word “significantly” is a pleonasm; just use one, not both; correct accordingly along with the whole manuscript as well as Tables; this is a must. We previously requested to eliminate the suffix (r=xxxx), and the word “P-value” from the Correlation Tables; yet, the authors did not consider our request. The authors need to understand that it makes no sense at all to repeat such information along with all the correlated response variables, just include in the superior part the correlation value and above the P-value in parenthesis (i.e. 0.045); the use of such repeated word & suffixes only creates visual contamination. Regarding the Discussion section, it is quite long; it must be shortened. Besides, as already mentioned, the authors must link, in a logical fashion, their main findings along with the discussion section, to compare & to discuss and, thereafter, be able to propose some physiologic explanations for such specific outcomes, considering to previous similar studies from the scientific literature. In general, the authors made an accurate interpretation of the main findings; the authors must focus their main findings and confront them with respect to the scientific literature in a logical and focused fashion.  Some information is repetitive and must be eliminated; the Discussion section must be shortened and focused in a logical and scientific fashion. The Discussion must be precise, honest, fair, and in accordance with the obtained results and the main objectives of the study. The list of references cited in the manuscript is proper, yet, is quite extended; I strongly suggest eliminating all the bibliography older than the year 2000, also, 60% of the citations must be within the years 2015-2021; this is a must. The authors did not work on this request.  The authors need to complete some improvements both the English language quality as well as the clarity and logical arrangement of the observed results, especially in the Discussion section and the Conclusion section; not always more is better.  The whole manuscript, especially the Introduction, Results, and Discussion sections must be significantly reduced. At this point, and based on the above comments, my pronouncement is that this manuscript could be accepted after moderate revisions. 

Author Response

To Reviewer: 1

We are most grateful to Reviewer 1 for the critical comments and useful suggestions that have helped us to improve our paper considerably.  As indicated in the responses that follow, we have taken all these comments and suggestions into account in the revised version of our paper.  The changes to the text have been indicated by the track changes mode in the revised manuscript.  We sincerely hope the revised manuscript meets your comments.

8 Your comment:

The results section of the abstract is very long and particularly confusing; I do recommend including in one “block” all the positive correlations, and then, all the negative correlations, but only those denoting significant correlations, either positive or negative. As previously suggested, the authors must select some key variables to include the observed value at least in the best experimental group along with the probability level. In this new R1 version, the authors use the “positive approach” regarding the excellent compensatory responses observed in the PREG group to face such a physiologic stage.  

 8 Our response:

Thank you for your comment. As you pointed out in your comment, we revised the manuscript giving priority to the positive correlations, and then, the negative correlations.

---------------------------------------------------------------------------------------------------------------------

8 Your comment:

As previously suggested, the authors must select some key variables to include the observed value at least in the best experimental group along with the probability level.

8 Our response:

Thank you for your comment. As you pointed out in your comment, the authors must select some key variables to include the observed value at least in the best experimental group along with the probability level as following;

We used many items that described the changes associated with pregnancy and lactation in she-camels that included steroid hormones, lipid peroxidation variables, antioxidant indicators and lipid profiles/liver functions biomarkers. We hope that these items presented clear descriptions of the clinical status as well as stress associated different stages of production and reproduction of she-camels.

--------------------------------------------------------------------------------------------------------------------

8 Your comment:

The abstract sections MUST be shortened.

8 Our response:

Thank you for your comment. We revised the abstract section and now it became shorter. However, the study contained more than 20 parameters covering different stages of production and reproduction in she-camels that required enough clarification. In our humble opinion, we think that more reduction may negatively affect the concept and value of the current study.

--------------------------------------------------------------------------------------------------------------------

8 Your comment:

It was previously mentioned that is important to include the temperature-humidity index (THI) observed along with the experimental period, to better relate the physiologic responses of females to their environmental context.

8 Our response:

Thank you for your comment. We revised the materials and methods as recommended and added one more figure (Figure 2) and this additional paragraph as follow;

Alexandria weather data (December to May) including maximum temperature (T, °C), and relative humidity (RH, %) were used to determine the temperature-humidity index (THI) (Fig. 2) using the following equation [37]:

The threshold value of heat stress in she-camels was set as 72-points THI, where the stress was categorized into mild (72-79 THI), moderate (80-89 THI), and severe stress (≥90 THI) [37].

Fig. 2. The temprature-humidity index (THI) and the weather data in Alexandria during the experimental period of the study (December to May). The figure is showing the weather data included relative humidity (%; black line), maximum temprature (°C; violet line), and THI (gradient column). The threshold level for mild stress (72-79 THI) is shown as squar dot line. The values inside each column refers to the THI value during this month.

--------------------------------------------------------------------------------------------------------------------

8 Your comment:

No Simple Summary was included.

8 Our response:

Thank you for your comment. As you pointed out in your comment, that no Simple Summary was included as following;

The manuscript included an abstract “in which we provided a summarized and concise description of the reported data” and conclusions according to the instruction of the journal that does not require summary.

---------------------------------------------------------------------------------------------------------------------

8 Your comment:

In the Introduction section, every statement included at least one reference; this section is quite long; it must be shortened. Some information must be moved to the Discussion section.

8 Our response:

Thank you for your comment. As you pointed out in your comment, our response as following;

We provided the requested references. And the introduction was revised. We hope these revisions meets the reviewer comments.

---------------------------------------------------------------------------------------------------------------------

8 Your comment:

Again, no working hypothesis of the study was proposed; this is a must.

8 Our response:

Thank you for your comment. As you pointed out in your comment, our response as following;

working hypothesis of the study was revised as following;

Up to our knowledge, there limited data was available regarding the relation between steroid hormones, biochemical parameters, oxidant and antioxidants parameters and both pregnancy and lactation in Dromedary she-camels. The current study aimed to assess the efficacy of controlled-internal drug releasing and its relationship with conception rates and clinical changes one-humped she-camels. It also focused on the changes in steroid hormones i.e. P4, E2 and cortisol, oxidant/antioxidants biomarkers i.e. MDA as an indicator of lipid peroxidation and oxidative stress, and SOD, TAC, CAT and GSH as anti-oxidants, lipid profiles/liver functions indices, renal function biomarkers and minerals metabolism indicators in non-pregnant, lactating or pregnant one-humped camels (Camelus dromedarius) pre-synchronized by controlled-internal drug releasing. The study also focused on the correlation relationships between steroid hormones and each of oxidant/antioxidant biomarkers, lipid profiles and liver functions indices, renal functions and mineral metabolism in these she-camels. . Based on the overmentioned background and the aim of the study our experiment hy-pothesis is that different pregnancy and lactation status affecting the level of steroid hor-mones, oxidants/antioxidant bio-markers, liver functions and kidney functions bio-markers, and mineral metabolism, in CIDR pre-synchronized she-camels.

---------------------------------------------------------------------------------------------------------------------

8 Your comment:

However, whereas the authors did not explain why they are reporting means instead of least-square means, they also never mentioned why they used Pearson´s Correlation Analyses instead of the Spearman Correlation procedure for some defined variables.  

8 Our response:

Thank you for your comment. As you pointed out in your comment, our response as following; We used both of reporting means and used Pearson´s Correlation Analyses instead of least-square means and the Spearman Correlation procedure for some defined variables, respectively, because these were more appropriate to the data of measured parameters in the current study as Pearson correlation evaluates the linear relationship between two continuous variables when the data is normally distribute.

-------------------------------------------------------------------------------------------------------------------------------

8 Your comment:

The authors must include some kind of quantitative information aside from the P-values of the observed results regarding the response variables.  Also, if no differences occurred among experimental groups for a defined variable, the authors must include the average value for such response variables observed in the study

8 Our response:

Thank you for your comment. As you pointed out in your comment, our response as following; regarding to quantitative information aside from the P-values of the observed results regarding the response variables.  Also, if no differences occurred among experimental groups for a defined variable, the authors must include the average value for such response variables observed in the study,

all these values were provided within the associated tables throughout the study. Insertion of these values within the text is some sort of unnecessary repetition. All internal journals ask not to repeat values if they were mentioned in tables.

-------------------------------------------------------------------------------------------------------------------------------

8 Your comment:

Further, as previously mentioned, the inclusion of the “p-values” along with the word “significantly” is a pleonasm; just use one, not both; correct accordingly along with the whole manuscript as well as Tables; this is a must.

8 Our response:

Thank you for your comment. As you pointed out in your comment, our response as following;

We deleted “p-values and we used the word “significantly” as recommended.

-------------------------------------------------------------------------------------------------------------------

8 Your comment:

We previously requested to eliminate the suffix (r=xxxx), and the word “P-value” from the Correlation Tables; yet, the authors did not consider our request. The authors need to understand that it makes no sense at all to repeat such information along with all the correlated response variables, just include in the superior part the correlation value and above the P-value in parenthesis (i.e. 0.045); the use of such repeated word & suffixes only creates visual contamination.

8 Our response:

Thank you for your comment. As you pointed out in your comment, our response as following; We eliminated the suffix (r=xxxx), and the word “P-value” from the Correlation Tables. We also included in the superior part the correlation value and above the P-value in parenthesis.

-------------------------------------------------------------------------------------------------------------------------------

8 Your comment:

Regarding the Discussion section, it is quite long; it must be shortened. Besides, as already mentioned, the authors must link, in a logical fashion, their main findings along with the discussion section, to compare & to discuss and, thereafter, be able to propose some physiologic explanations for such specific outcomes, considering to previous similar studies from the scientific literature.

8 Our response:

Thank you for your comment. As you pointed out in your comment, our response as following; We have revised the discussion section as recommended and deleted the repeated parts. We hope these revisions meets the reviewer recommendation. We tried to clarify the most important findings in investigated she-camels. This necessitated full description for more than 20 biochemical parameters as well as clinical findings and steroids hormones in different stages of production and reproduction.

8 Your comment:

The list of references cited in the manuscript is proper, yet, is quite extended; I strongly suggest eliminating all the bibliography older than the year 2000, also, 60% of the citations must be within the years 2015-2021; this is a must. The authors did not work on this request.

8 Our response:

Thank you for your comment. As you pointed out in your comment, our response as following;

We removed 32 reference from the references list, added one more recent reference “Habte et al. 2021” (covering the Temperature-humidity index in camel) and revised the order of all references list. Most of the references are recent as the reviewer strongly recommend. We would like to explain that we had to include some references which included the reference(s) value(s) of the measured parameters in camels.

Thank you so much for your valuable comments that helped us to improve our manuscript. We hope the current revised version of the manuscripts meets all your valuable comments.
